# Scrutinize What We Ignore: Reining In Task Representation Shift Of Context-Based Offline Meta Reinforcement Learning

**Hai Zhang**[1], **Boyuan Zheng**[1], **Tianying Ji**[2], **Jinhang Liu**[1], **Anqi Guo**[1]
**Junqiao Zhao**[1]\*, **Lanqing Li**[3,4]\*

[1] Tongji University, [2] Tsinghua University, [3] Zhejiang Lab, [4] The Chinese University of HongKong

```
{zhanghai12138, zhengboyuan, jinhangliu, zhaojunqiao}@tongji.edu.cn
jity20@mails.tsinghua.edu.cn
{anqiguo098, lanqingli1993}@gmail.com
```

## Abstract

Offline meta reinforcement learning (OMRL) has emerged as a promising approach for interaction avoidance and strong generalization performance by leveraging pre-collected data and meta-learning techniques. Previous context-based approaches predominantly rely on the intuition that alternating optimization between the context encoder and the policy can lead to performance improvements, as long as the context encoder follows the principle of maximizing the mutual information between the task variable $M$ and its latent representation $Z$ ($I(Z; M)$) while the policy adopts the standard offline reinforcement learning (RL) algorithms conditioning on the learned task representation. Despite promising results, the theoretical justification of performance improvements for such intuition remains underexplored. Inspired by the return discrepancy scheme in the model-based RL field, we find that the previous optimization framework can be linked with the general RL objective of maximizing the expected return, thereby explaining performance improvements. Furthermore, after scrutinizing this optimization framework, we observe that the condition for monotonic performance improvements does not consider the variation of the task representation. When these variations are considered, the previously established condition may no longer be sufficient to ensure monotonicity, thereby impairing the optimization process. We name this issue task representation shift and theoretically prove that the monotonic performance improvements can be guaranteed with appropriate context encoder updates. We use different settings to rein in the task representation shift on three widely adopted training objectives concerning maximizing $I(Z; M)$ across different data qualities. Empirical results show that reining in the task representation shift can indeed improve performance. Our work opens up a new avenue for OMRL, leading to a better understanding between the task representation and performance improvements. [1]

## 1 Introduction

RL has driven impressive advances in many complex decision-making problems in recent years (Silver et al., 2018; Schrittwieser et al., 2020; Zhang et al., 2023b; 2024; Ma et al., 2024b;a), primarily through online RL methods. However, the extensive interactions required by online RL entail high costs and safety concerns, posing significant challenges for real-world applications. Offline RL (Wu et al., 2019; Levine et al., 2020) offers an appealing alternative by efficiently leveraging pre-collected data for policy learning, thereby circumventing the need for online interaction with the environment. This advantage extends the application of RL, covering healthcare (Fatemi et al., 2022; Tang et al., 2022), robotics (Sinha et al., 2022; Kumar et al., 2022) and games (Schrittwieser et al., 2021).

---

\*Corresponding Author
[1]Code: https://github.com/betray12138/Task-Representation-Shift

Though demonstrating its superiority, offline RL holds a notable drawback towards generalizing to the unknown (Ghosh et al., 2021).

As a remedy, by combining the meta-learning techniques, OMRL (Li et al., 2020; Mitchell et al., 2021; Xu et al., 2022) has emerged as an effective training scheme toward strong generalization performance and fast adaptation capability while maintaining the merits of offline RL. Among the OMRL research, context-based OMRL (COMRL) algorithms (Li et al., 2020; 2024) hold a popular paradigm that seeks optimal meta-policy conditioning on the context of Markov Decision Processes (MDPs). Specifically, these methods (Li et al., 2020; Gao et al., 2024; Li et al., 2024) propose to train a context encoder via maximizing $I(Z; M)$ to learn the task representation from the collected context (*e.g. trajectories*) and train the downstream policy with standard offline RL algorithms conditioning on the task representation, as shown in Figure 1 left. The context encoder and policy are trained in an alternating manner, with each being updated once per cycle. Despite promising results, the theoretical justification of performance improvements for such intuition remains underexplored.

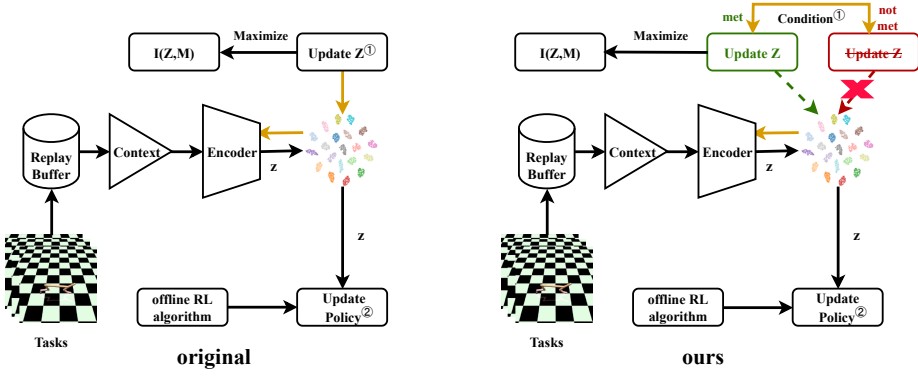

Figure 1: **Our training framework compared to the previous training framework.** They both adopt the alternating optimization framework to train the context encoder and the policy. However, our training framework considers the previously ignored variation of task representation by introducing an extra condition to decide whether the context encoder should be updated.

Intriguingly, we formalize this training framework as an alternating two-stage optimization framework and then link it with the general RL objective of maximizing the expected return. To be specific, maximizing $I(Z; M)$ and adopting standard offline RL algorithms can be interpreted as consistently raising the lower bound of the expected return conditioning on the optimal task representation distribution (See Section 4.1). This is achieved by extending the return discrepancy scheme (Janner et al., 2019) to the COMRL framework. Thus, this analysis provides a feasible explanation for the performance improvement guarantee.

More importantly, after scrutinizing this optimization framework, we find it ignores the impacts stemming from the variation of the task representation in the alternating process. This would cause the optimization framework to incorrectly conclude that the monotonic performance improvement can be guaranteed with a better approximation to the optimal task representation distribution. By explicitly modeling the variation of task representation, we conclude that it is a critical part of monotonic performance improvement. To highlight the characteristic, we name this issue task representation shift and theoretically prove that it is possible to achieve monotonic performance improvement with appropriate context encoder updates (See Section 4.2).

To show the impacts of this issue, we use different settings to rein in the task representation shift on three widely adopted objectives concerning maximizing $I(Z; M)$, covering the upper bound, the lower bound, and the direct approximation. Empirical results show that reining in the task representation shift can indeed improve performance. Our work opens up a new avenue for OMRL, leading to a better understanding between the task representation and performance improvements.

## 2 RELATED WORKS

**Context-Based Offline Meta RL.** As the marriage between context-based meta RL and offline RL, COMRL combines the merits of both sides. Specifically, COMRL methods (Li et al., 2020; Yuan & Lu, 2022; Gao et al., 2024; Li et al., 2024) leverage the offline dataset to train a context encoder to learn robust task representations, and then pass the representation to the policy and value function as input. At test time, COMRL methods leverage the generalization ability of the learned context encoder to perform meta-adaptation. According to the theoretical insights from (Li et al., 2024), prior COMRL works (Li et al., 2020; Yuan & Lu, 2022; Gao et al., 2024; Li et al., 2024) mainly focus on designing a context encoder learning algorithm to better approximate $I(Z; M)$. However, our work turns the focus to refining the condition of monotonic performance improvements based on a given context encoder learning algorithm. Centering around this motivation, we identify a new issue called task representation shift, which is ignored by the previous COMRL endeavors.

**Performance Improvement Guarantee.** Ensuring performance improvement is a key concern in both online and offline reinforcement learning settings. In online RL, performance improvement guarantees are often established through methods such as performance difference bounds (Kakade & Langford, 2002; Schulman et al., 2015; Ji et al., 2022; Zhang et al., 2023a), return discrepancy (Luo et al., 2018; Janner et al., 2019), and regret bounds (Osband & Van Roy, 2014; Curi et al., 2020). In offline RL, CQL-based methods (Kumar et al., 2020; Yu et al., 2021) also enjoy safe policy improvement guarantees. However, most works focus on the single-task setting, leaving the performance improvement guarantees in the context-based meta-RL settings largely unexplored. While ContrBAR (Choshen & Tamar, 2023) also benefits from the performance improvement guarantee, the theoretical insight focuses on the online setting. Additionally, it is tailored to one particular approximation of $I(Z; M)$ as it makes assumptions specific to this approximation. In contrast, our work focuses on the general offline setting, addressing a broader class of algorithms that maximize various bounds of $I(Z; M)$.

## 3 PRELIMINARIES

### 3.1 PROBLEM STATEMENT

A task in RL is generally formalized as a fully observed MDP (Puterman, 2014), which is represented by a tuple $(\mathcal{S}, \mathcal{A}, P, \rho_0, R, \gamma)$ with state space $\mathcal{S}$, action space $\mathcal{A}$, transition function $P(s'|s, a)$, reward function $R(s, a)$, initial state distribution $\rho_0(s)$ and discount factor $\gamma \in [0, 1]$. For arbitrary policy $\pi(a|s)$, we denote the state-action distribution at timestep $t$ as $d_{\pi,t}(s, a) \triangleq \Pr(s_t = s, a_t = a|s_0 \sim \rho_0, a_t \sim \pi, s_{t+1} \sim P, \forall t \geq 0)$. The discounted state-action distribution of given $\pi(a|s)$ is denoted as $d_\pi(s, a) \triangleq (1 - \gamma) \sum_{t=0}^{\infty} \gamma^t d_{\pi,t}(s, a)$. The ultimate goal is to find an optimal policy $\pi(a|s)$ to maximize the expected return $\mathbb{E}_{(s,a) \sim d_\pi}[R(s, a)]$.

It is widely assumed that the task $m$ in the meta RL setting is randomly sampled from the task distribution $p(M)$ (Li et al., 2020; Yuan & Lu, 2022; Gao et al., 2024; Li et al., 2024). We denote the number of training tasks as $N$. For each task $i \in [0, 1, ..., N - 1]$, an offline dataset $D_i = \{(s_{i,j}, a_{i,j}, s'_{i,j}, r_{i,j})\}_{j=1}^{K}$ is collected in advance, where $K$ denotes the number of transitions. The learning algorithm is required to train a meta-policy $\pi_{\text{meta}}$ with access only to the given offline datasets. At test time, given an unseen task, the meta policy $\pi_{\text{meta}}$ performs task adaptation to get a task-specific policy and then will be evaluated in the environment.

### 3.2 CONTEXT-BASED OMRL

Previous COMRL endeavors employ a compact representation $z$ to capture/quantify the variation over tasks (Yuan & Lu, 2022). In practice, they choose to train a context encoder $Z(\cdot|x; \phi)$ to extract the task information (Li et al., 2020; Dorfman et al., 2021), where $x$ denotes the given context and $\phi$ denotes the parameters of the context encoder. Then, the policy is learned by conditioning on the task representation as $\pi(a|s, Z(\cdot|x; \phi); \theta)$. We assume there exists an optimal task representation distribution $Z(\cdot|x; \phi^*)$ that meets the optimal expected return for any policy parameter $\theta$ is determined by $\pi(a|s, Z(\cdot|x; \phi^*); \theta)$. Hence, the objective of COMRL is formalized as:

$$\max_{\theta} J^*(\theta) = \mathbb{E}_{m,x}[\mathbb{E}_{(s,a) \sim d_{\pi(\cdot|s, Z(\cdot|x; \phi^*); \theta)}}[R_m(s, a)]] \tag{1}$$

where $R_m(s, a)$ denotes the ground-truth reward function of the task $m$.

Previous COMRL works (Li et al., 2020; 2021; Yuan & Lu, 2022; Gao et al., 2024; Li et al., 2024) are proven to optimize the context encoder by maximizing the approximate bounds of $I(Z; M)$, as shown in Theorem 3.1. They hold the intuition that using such kind of approach to optimize the context encoder and adopting the standard offline RL algorithms to optimize the policy can lead to performance improvements. However, the theoretical justification of performance improvement for such intuition has been less explored. For later use, we denote $Z(\cdot|x; \phi^{mutual})$ as the optimal solution for maximizing $I(Z; M)$. Whether $Z(\cdot|x; \phi^{mutual})$ is equivalent to $Z(\cdot|x; \phi^*)$ remains an open problem. To avoid confusion, we introduce an extra notation here.

**Theorem 3.1** ((Li et al., 2024)). *Denote $X_b$ and $X_t$ are the behavior-related $(s, a)$-component and task-related $(s', r)$-component of the context $X$, with $X = (X_b, X_t)$. We have:*

$$I(Z; X_t | X_b) \leq I(Z; M) \leq I(Z; X) \tag{2}$$

*where 1) $\mathcal{L}_{FOCAL} \equiv -I(Z; X) = -I(Z; X_t | X_b) - I(Z; X_b)$; 2) $\mathcal{L}_{CORRO} \equiv -I(Z; X_t | X_b)$; 3) $\mathcal{L}_{CSRO} \geq (\lambda - 1) I(Z; X) - \lambda I(Z; X_t | X_b)$, and $\equiv$ denotes equality up to a constant.*

## 4 METHODS

We first introduce a Lipschitz assumption, a widely used technique in both model-free (Song & Sun, 2019; Ghosh et al., 2022) and model-based (Ji et al., 2022; Zhang et al., 2023a) RL frameworks.

**Assumption 4.1.** *The policy function is $L_z-$ Lipschitz w.r.t some norm $|| \cdot ||$ in the sense that*

$$\forall Z(\cdot|x; \phi_1), Z(\cdot|x; \phi_2) \in \mathbb{Z}, \ |\pi(\cdot|s, Z(\cdot|x; \phi_1); \theta) - \pi(\cdot|s, Z(\cdot|x; \phi_2); \theta)| \leq L_z \cdot |Z(\cdot|x; \phi_1) - Z(\cdot|x; \phi_2)| \tag{3}$$

The proofs for all the following sections are provided in Appendix Section 8.2.

### 4.1 A PERFORMANCE IMPROVEMENT PERSPECTIVE TOWARDS PRIOR WORKS

Our goal is to arrive at an optimization objective for COMRL to get the performance improvement guarantee. Motivated by the return discrepancy scheme (Janner et al., 2019) in the model-based RL field, we can similarly build a tractable lower bound for $J^*(\theta)$.

**Definition 4.2** (Return discrepancy in COMRL). *The return discrepancy in the COMRL setting can be defined as*

$$J^*(\theta) - J(\theta) \geq -|J(\theta) - J^*(\theta)| \tag{4}$$

*where $J(\theta) = \mathbb{E}_{m,x}[\mathbb{E}_{(s,a) \sim d_{\pi(\cdot|s, Z(\cdot|x; \phi); \theta)}}[R_m(s, a)]]$ is the expected return of the policy $\pi$ conditioning on the learned task representation $Z(\cdot|x; \phi)$.*

From Definition 4.2, we know that if the estimation error $|J^*(\theta) - J(\theta)|$ can be upper-bounded, the lower bound for $J^*(\theta)$ can be established simultaneously. Then, Eq. (4) can be interpreted as a unified training objective for both the context encoder and the policy. Specifically, we can alternate the optimization target to lift the lower bound for $J^*(\theta)$, where the whole optimization process can be seen as a two-stage alternating framework with the first stage updating the context encoder to minimize $|J^*(\theta) - J(\theta)|$ and the second stage updating the policy to maximize $J(\theta) - |J^*(\theta) - J(\theta)|$ as $J^*(\theta) \geq J(\theta) - |J^*(\theta) - J(\theta)|$. Hence, the performance improvement guarantee can be achieved.

The following theorem induces a tractable upper bound for $|J^*(\theta) - J(\theta)|$, bridging the gap between the intuition of previous works and the theoretical justification concerning the performance improvement guarantee.

**Theorem 4.3** (Return bound in COMRL). *Assume the reward function is upper-bounded by $R_{max}$. For an arbitrary policy parameter $\theta$, when meeting Assumption 4.1, the return bound in COMRL can be formalized as:*

$$|J^*(\theta) - J(\theta)| \leq \frac{2R_{max}L_z}{(1-\gamma)^2} \mathbb{E}_{m,x}(|Z(\cdot|x; \phi) - Z(\cdot|x; \phi^{mutual})| + |Z(\cdot|x; \phi^{mutual}) - Z(\cdot|x; \phi^*)|) \tag{5}$$

By directly unrolling the Theorem 4.3, we can establish the lower bound of $J^*(\theta)$ as:

$$J^*(\theta) \geq J(\theta) - \frac{2R_{max}L_z}{(1-\gamma)^2}\mathbb{E}_{m,x}(|Z(\cdot|x;\phi) - Z(\cdot|x;\phi^{mutual})| + |Z(\cdot|x;\phi^{mutual}) - Z(\cdot|x;\phi^*)|)$$

(6)

Note that $|Z(\cdot|x;\phi^{mutual}) - Z(\cdot|x;\phi^*)|$ is a constant w.r.t the learned task representation $Z(\cdot|x;\phi)$. Hence, this term can be ignored when optimizing $Z(\cdot|x;\phi)$. Admittedly, there may exist a smarter algorithm to further reduce the gap of $|Z(\cdot|x;\phi^{mutual}) - Z(\cdot|x;\phi^*)|$, but this is not our main focus and we leave this to future work.

Theorem 4.3 and Eq. (6) indicate the following two principles for the context encoder and the policy. For the context encoder learning, we should minimize the return bound, namely minimizing $\mathbb{E}_{m,x}|Z(\cdot|x;\phi) - Z(\cdot|x;\phi^{mutual})|$. For the policy learning, we should lift the lower bound of $J^*(\theta)$, namely maximizing $J(\theta) - \frac{2R_{max}L_z}{(1-\gamma)^2}\mathbb{E}_{m,x}(|Z(\cdot|x;\phi) - Z(\cdot|x;\phi^{mutual})| + |Z(\cdot|x;\phi^{mutual}) - Z(\cdot|x;\phi^*)|) \equiv J(\theta)$, as the task representation related terms are constants for policy optimization. To maximize $J(\theta)$, we can adopt standard offline RL algorithms (Wu et al., 2019; Kumar et al., 2020; Fujimoto & Gu, 2021) similar to (Yang et al., 2022).

Recall that previous COMRL works adopt alternatingly training the context encoder by maximizing $I(Z; M)$, which can be approximately seen as minimizing $\mathbb{E}_{m,x}|Z(\cdot|x;\phi) - Z(\cdot|x;\phi^{mutual})|$ and training the policy by conditioning on the learned task representation as well as applying the standard offline RL algorithms. Therefore, we argue that our theoretical analyses provide an explanation for the performance improvement guarantee of previous COMRL works.

However, this optimization framework only considers the discrepancy between $Z(\cdot|x;\phi)$ and $Z(\cdot|x;\phi^{mutual})$, without considering the variation of $Z(\cdot|x;\phi)$. In the next section, we will show that this may violate the monotonicity of the performance improvements.

## 4.2 MONOTONIC PERFORMANCE IMPROVEMENT CONCERNING TASK REPRESENTATION SHIFT

In this section, we aim to demonstrate that as the previous optimization framework in Section 4.1 doesn't model the variation of task representation explicitly, this framework would provide an insufficient condition for monotonic performance improvement. We first show conditions for monotonic performance improvement of the previous framework.

**Corollary 4.4** (Monotonic performance improvement condition for previous COMRL works). *When meeting Assumption 4.1, the condition for monotonic performance improvement of previous COMRL works is:*

$$\epsilon_{12}^* \triangleq J^*(\theta_2) - J^*(\theta_1) \geq \frac{4R_{max}L_z}{(1-\gamma)^2}\mathbb{E}_{m,x}(|Z(\cdot|x;\phi) - Z(\cdot|x;\phi^*)|)$$

(7)

As shown in Corollary 4.4, $Z(\cdot|x;\phi)$ should be close to $Z(\cdot|x;\phi^*)$ such that the lower bound is small enough for finding a policy to achieve monotonic performance improvement. Next, we will introduce the performance difference bound framework to model variation of task representation.

**Definition 4.5** (Performance difference bound in COMRL). *For an alternating update process, we denote $J^1(\theta_1) = \mathbb{E}_{m,x}\mathbb{E}_{(s,a)\sim d_{\pi(\cdot|s,Z(\cdot|x;\phi_1);\theta_1)}}[R_m(s,a)]$ as the expected return of the policy $\pi(\cdot|s, Z(\cdot|x;\phi_1); \theta_1)$ before update of the context encoder and the policy. Similarly, denote $J^2(\theta_2) = \mathbb{E}_{m,x}\mathbb{E}_{(s,a)\sim d_{\pi(\cdot|s,Z(\cdot|x;\phi_2);\theta_2)}}[R_m(s,a)]$ as the expected return of the policy $\pi(\cdot|s, Z(\cdot|x;\phi_2); \theta_2)$ after update of the context encoder and the policy. The performance difference bound in the COMRL setting can be defined as*

$$J^2(\theta_2) - J^1(\theta_1) \geq C$$

(8)

*when $C$ is non-negative, the algorithm allows a monotonic performance improvement.*

According to Definition 4.5, we need to find a positive $C$ to improve the performance monotonically. To achieve this, we can derive the lower bound of the performance difference.

**Theorem 4.6** (Lower bound of performance difference in COMRL). *Assume the reward function is upper-bounded by $R_{max}$. When meeting Assumption 4.1, the lower bound of the performance difference in COMRL can be formalized as:*

$$J^2(\theta_2) - J^1(\theta_1) \geq \epsilon_{12}^* - \frac{2R_{max}L_z}{(1-\gamma)^2}\mathbb{E}_{m,x}[2|Z(\cdot|x;\phi_2) - Z(\cdot|x;\phi^*)| + |Z(\cdot|x;\phi_2) - Z(\cdot|x;\phi_1)|] \tag{9}$$

For achieving the monotonic performance improvement guarantee, we further let the right-hand side of Eq. (9) be positive. Then, we would have the following condition to achieve monotonicity.

$$\epsilon_{12}^* - \frac{2R_{max}L_z}{(1-\gamma)^2}\mathbb{E}_{m,x}[2|Z(\cdot|x;\phi_2) - Z(\cdot|x;\phi^*)| + |Z(\cdot|x;\phi_2) - Z(\cdot|x;\phi_1)|] \geq 0 \tag{10}$$

Compared to the condition in Corollary 4.4, Eq. 10 presents an additional term $|Z(\cdot|x;\phi_2) - Z(\cdot|x;\phi_1)|$ to achieve monotonic performance improvement, which corresponds exactly to the previous ignored impacts stemming from the variation of the task representation. To achieve monotonic performance improvements, the optimization of task representation should consider not only the approximation to $Z(\cdot|x;\phi^*)$, but also the magnitude of the update. For example, if we assume that $Z(\cdot|x;\phi_1)$ is trained from scratch and the update process from $Z(\cdot|x;\phi_1)$ to $Z(\cdot|x;\phi_2)$ brings $Z(\cdot|x;\phi_2)$ close to $Z(\cdot|x;\phi^*)$, then with the condition in Corollary 4.4, the monotonic performance improvement can be easily achieved with small $\epsilon_{12}^*$. However, for the condition in Eq. (10), small $\epsilon_{12}^*$ may cause the violation of monotonicity as $|Z(\cdot|x;\phi_1) - Z(\cdot|x;\phi_2)|$ remains large.

To highlight the impacts of $|Z(\cdot|x;\phi_2) - Z(\cdot|x;\phi_1)|$, we name this issue Task Representation Shift. Under some mild assumptions, we can conclude that it is possible to achieve monotonic performance improvement with sufficient policy improvement $\epsilon_{12}^*$ on the condition of appropriate encoder updates, as shown in Theorem 4.10.

**Assumption 4.7.** *The impacts of task representation shift are upper-bounded by $\beta$ and less than the policy improvement $\epsilon_{12}^*$ with a certain coefficient.*

**Assumption 4.8.** *The space of the task representation is discrete and limited.*

**Assumption 4.9.** *There exists an $\alpha$ for any given $Z(\cdot|x;\phi_1)$, $|Z(\cdot|x;\phi_2) - Z(\cdot|x;\tilde{\phi}_2)|^2 \leq \frac{\alpha}{b}$, where $Z(\cdot|x;\phi_2)$ denotes the context encoder updated by maximizing $I(Z;M)$ with the data size $b$ randomly sampled from the training dataset based on $Z(\cdot|x;\phi_1)$ and $Z(\cdot|x;\tilde{\phi}_2)$ denotes the context encoder updated by fitting the empirical distribution on $b$ i.i.d samples from $Z(\cdot|x;\phi^*)$ based on $Z(\cdot|x;\phi_1)$ [2].*

**Theorem 4.10** (Monotonic performance improvement guarantee on training process). *Denote $\kappa$ as $\frac{(1-\gamma)^2}{4R_{max}L_z}\epsilon_{12}^* - \frac{1}{2}\beta$ and $|Z|$ as the cardinality of the task representation space. Given that the context encoder has already been trained by maximizing $I(Z;M)$ to some extent. When meeting Assumption 4.1, 4.7, 4.8 and 4.9, with a probability greater than $1 - \xi$, we can get the monotonic performance improvement guarantee by updating the context encoder via maximizing $I(Z;M)$ from at least extra $k$ samples, where:*

$$k = \frac{1}{\kappa^2}(\sqrt{2\log\frac{2^{|Z|} - 2}{\xi}} + \sqrt{\alpha})^2 \tag{11}$$

*Here, $\xi \in [0,1]$ is a constant.*

Theorem 4.10 shows the connection between the needed update data size $k$ and the performance improvement brought only by the policy update $\epsilon_{12}^*$. If the calculated $k$ is larger than the given batch size, the context encoder should avoid this update and wait for the accumulation of $\epsilon_{12}^*$. As $\epsilon_{12}^*$ increases, $k$ decreases. When $k$ is smaller than the given batch size, the monotonic performance improvement can be achieved by updating the context encoder. Notice that this updating process can be performed multiple times, as long as satisfying Assumption 4.7. The updating process is visualized in Figure 1 right. Compared to the original alternating framework, Theorem 4.10 unveils that the core part for better performance improvement is to adjust the update of the context encoder to rein in the task representation shift, thereby showcasing the advantage over the previous works. The general algorithmic framework is shown in Algorithm 1, where the way to adjust the update of the context encoder is colored by red.

---

[2]For more details to justify these assumptions, please refer to Appendix 8.3.

---

**Algorithm 1** General Algorithmic Framework Towards Reining In The Task Representation Shift

---

**Input:** Offline training datasets $X$, initialized policy $\pi_\theta$, context encoder $Z_\phi$, given task representation shift threshold $\beta$ and given training batch size for context encoder $N_{bs}$.

1: **for** iter in alternating iterations **do**
2:     // Update the context encoder
3:     Estimate the conditions (e.g.use Eq. (11) to approximate $k$)
4:     **if** Conditions for updating the context encoder are met (e.g. $k <= N_{bs}$) **then**
5:         **while** Accumulated task representation shift is less than $\beta$ **do**
6:             Sample context from $X$
7:             Obtain task representations from $Z_\phi$ with inputting the sampled context
8:             Compute $\mathcal{L}_{encoder}$ concerning maximizing $I(Z; M)$
9:             Update $\phi$ to minimize $\mathcal{L}_{encoder}$
10:         **end while**
11:     **end if**
12:     // Update the policy
13:     Detach the task representations
14:     Sample training data from $X$
15:     Compute $\mathcal{L}_{policy}$ via standard offline RL algorithms
16:     Update $\theta$ to minimize $\mathcal{L}_{policy}$
17: **end for**

---

### 4.3 PRACTICAL IMPLEMENTATION

As outlined in Algorithm 1, the way to rein in the task representation shift can be seen as two aspects, namely 1) when to update the context encoder determined by $k$ and 2) how many times to update the context encoder determined by $\beta$. To cover these two aspects, we introduce two parameters $N_k$ and $N_{acc}$. Here, $N_k = n$ denotes that the context encoder needs to be updated every $n$ updates of the policy, and $N_{acc} = n$ denotes when the context encoder needs to be updated, it is updated $n$ times. Our settings include 1) $N_k = 2, N_{acc} = 1$, 2) $N_k = 3, N_{acc} = 1$, 3) $N_k = 1, N_{acc} = 2$, and 4) $N_k = 1, N_{acc} = 3$. All settings as well as the original setting, namely $N_k = 1, N_{acc} = 1$, are performed for 8 different random seeds.

With respect to the specific context encoder learning algorithms concerning maximizing $I(Z; M)$, we choose three representative algorithms that have been widely adopted in previous COMRL endeavors. The details of these algorithms are shown in Appendix Section 8.4.

**Contrastive-based** is applied in (Li et al., 2020; Gao et al., 2024; Li et al., 2024) and it is proven to be the upper bound of $I(Z; M)$.

**Reconstruction-based** is applied in (Zintgraf et al., 2019; Dorfman et al., 2021; Li et al., 2024) and it is proven to be the lower bound of $I(Z; M)$, which is equivalent to CORRO (Yuan & Lu, 2022) under the offline setting.

**Cross-entropy-based** is proposed in our work and it is a direct approximation w.r.t $I(Z; M)$.

With respect to the policy learning algorithm, to maintain the consistency of previous COMRL works, we directly adopt BRAC (Wu et al., 2019) to train the policy.

## 5 EXPERIMENT

Our experiments are conducted to show that the previous optimization framework that ignores the impacts of task representation shift is not sufficient. We hope to illustrate the potential of reining in the task representation shift, laying the foundation for further research towards better performance improvement of COMRL.

### 5.1 ENVIRONMENTS SETTINGS

We adopt MuJoCo (Todorov et al., 2012) and MetaWorld (Yu et al., 2020) benchmarks to evaluate the algorithms. Following the protocol of UNICORN (Li et al., 2024), we randomly sample 20

training tasks and 20 testing tasks from the task distribution. We train the SAC (Haarnoja et al., 2018) agent from scratch for each task and use the collected replay buffer as the offline dataset. During the meta-testing phase, we adopt a fully offline setting in a few-shot manner like (Li et al., 2020) that randomly samples a trajectory from the dataset as the context to obtain the task representation and then passes it to the meta-policy to complete the testing in the true environments.

## 5.2 MAIN RESULTS

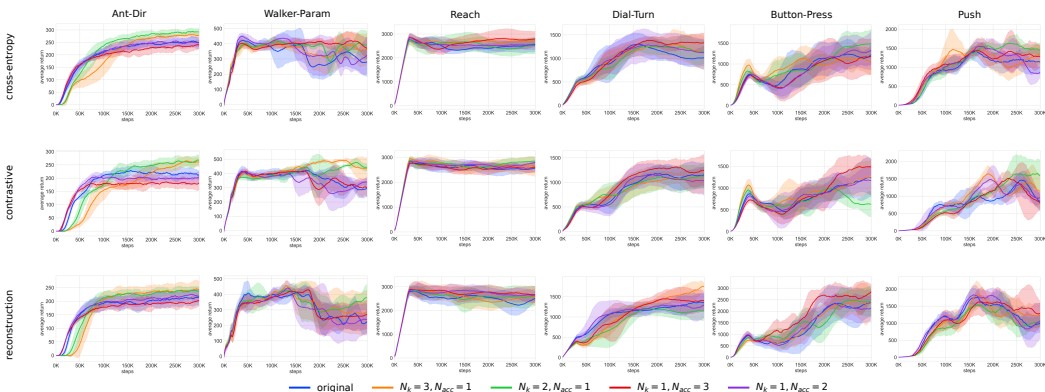

Figure 2: **Testing returns of different settings to rein in the task representation shift.** Solid curves refer to the mean performance of trials over 8 random seeds, and the shaded areas characterize the standard deviation of these trials.

Figure 2 and Table 3 illustrate the performance of our different settings to rein in the task representation shift on 20 testing tasks. We find that these specific objectives exhibit clear trends across all benchmarks that the best setting showcases statistically significant performance improvements compared to the worst setting, thereby emphasizing the indispensability of considering the task representation shift issue. We also notice that the previous training framework, namely $N_k = 1, N_{acc} = 1$, fails to deliver the best asymptotic performance across all benchmarks. Though $N_k = 1, N_{acc} = 1$ can be seen as an implicit way to rein in the task representation shift, the lack of explicit control still hinders the capability of these base algorithms to reach their full potential. Furthermore, as shown in Table 3, we observe that settings with $N_k > 1, N_{acc} = 1$ achieve better performance more frequently than those with $N_k = 1, N_{acc} > 1$ w.r.t the original setting $N_k = 1, N_{acc} = 1$. Based on this observation, we recommend prioritizing the adjustment of $N_k$ when tuning parameters. This not only tends to yield better performance but also offers the advantage of reducing training time. In contrast, increasing $N_{acc}$ would increase training time due to the need for multiple context encoder updates within a single alternating step.

## 5.3 CAN THE RESULTS SHOW CONSISTENCY ACROSS DIFFERENT DATA QUALITIES?

To make our claim more universal, we conduct the experiment on different data qualities. We collect three types of datasets with size being equal to the dataset used in Section 5.2. Depending on the quality of the behavior policies, we denote these datasets as random, medium, and expert respectively.

Figure 3 and Table 3 demonstrate the performance of our different settings to rein in the task representation shift on Ant-Dir. We find that the contrastive-based algorithm fails on the random dataset. Except for this case, the others show consistent results that reining in the task representation shift can lead to better performance improvements.

Based on these impressive results, we believe that developing a smarter algorithm to control the task representation shift automatically is an appealing direction and we leave this to future work.

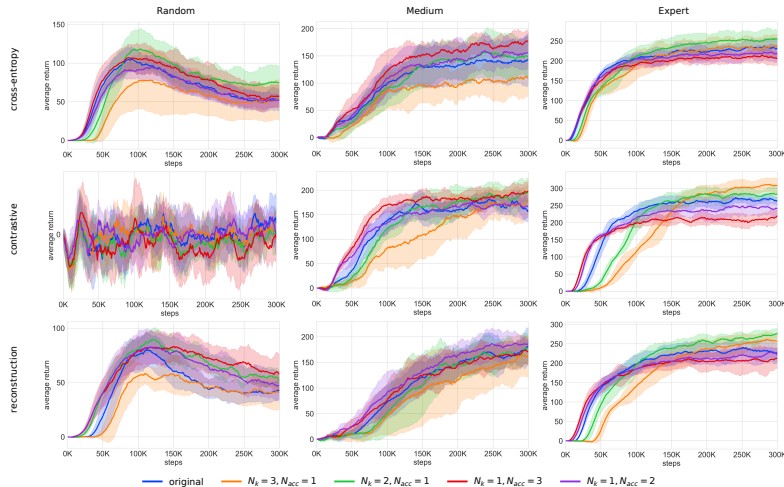

Figure 3: **Testing returns of different settings to rein in the task representation shift on different data qualities in Ant-Dir.** Solid curves refer to the mean performance of trials over 8 random seeds, and the shaded areas characterize the standard deviation of these trials.

# 6 DISCUSSION

This section seeks to raise some interesting issues derived from our work. We hope these issues can further enable pondering on the significance of the task representation shift.

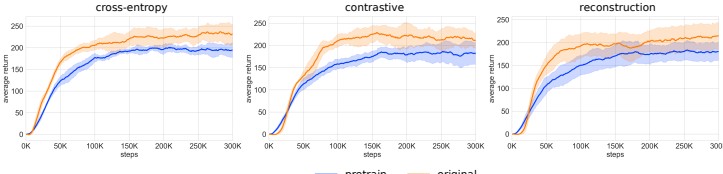

Figure 4: **Testing returns of the pre-training scheme against training from scratch in Ant-Dir.** Solid curves refer to the mean performance of trials over 8 random seeds, and the shaded areas characterize the standard deviation of these trials.

## 6.1 CAN THE PRETRAINING SCHEME BE ADOPTED TO ACHIEVE BETTER PERFORMANCE IMPROVEMENT?

According to our analysis, the task representation shift issue only happens in the case that the context encoder needs to be trained from scratch. Hence, a question is raised naturally. If we train the context encoder in advance and use this pre-trained context encoder to train the policy directly, can we achieve better performance improvement?

To answer this question, we conduct the experiment in Ant-Dir and still use the cross-entropy, contrastive, and reconstruction based objectives to pre-train the context encoder. As shown in Figure 4, there is a significant performance gap between pre-training and training from scratch. To better theoretically interpret this, we introduce the following Corollary.

**Corollary 6.1** (Monotonic performance improvement condition for pre-training scheme)**.** *Denote $Z(\cdot|x; \phi^{pretrain})$ as the task representation distribution after pre-training. When meeting Assumption 4.1, the monotonic performance improvement condition for pre-training scheme is:*

$$\epsilon_{12}^* - \frac{4R_{max}L_z}{(1-\gamma)^2} \mathbb{E}_{m,x}[|Z(\cdot|x; \phi^{pretrain}) - Z(\cdot|x; \phi^*)|] \geq 0 \quad (12)$$

While maximizing $I(Z; M)$ can help $Z(\cdot|x; \phi^{pretrain})$ to approach $Z(\cdot|x; \phi^{mutual})$, the sub-optimal gap between $Z(\cdot|x; \phi^{mutual})$ and $Z(\cdot|x; \phi^*)$ still persists. Furthermore, given the inherent approxima-

tion error between $Z(\cdot|x;\phi^{pretrain})$ and $Z(\cdot|x;\phi^{mutual})$, the discrepancy between $Z(\cdot|x;\phi^{pretrain})$ and $Z(\cdot|x;\phi^*)$ maybe large. Hence, according to Corollary 6.1, pre-training cannot actually achieve monotonic performance improvements. As the task representation cannot be changed under the pre-training scheme, compared to training from scratch, this scheme loses some degrees of freedom to improve performance.

Additionally, this scheme can be intuitively regarded as degrading to the problem that how to design a pre-training algorithm to better approximate $Z(\cdot|x;\phi^*)$. Notice that there exists the decomposition $|Z(\cdot|x;\phi^{pretrain}) - Z(\cdot|x;\phi^*)| \leq |Z(\cdot|x;\phi^{pretrain}) - Z(\cdot|x;\phi^{mutual})| + |Z(\cdot|x;\phi^{mutual}) - Z(\cdot|x;\phi^*)|$. Since the cross-entropy-based objective is the direct approximation of $I(Z;M)$, holding better performance under the pre-training scheme is also in line with expectation.

This analysis further enhances the importance of considering the impacts of task representation shift $|Z(\cdot|x;\phi_2) - Z(\cdot|x;\phi_1)|$ in the alternating process. Nevertheless, how to achieve better performance under the pre-training scheme is also an interesting problem.

## 6.2 CAN THE VISUALIZATION OF TASK REPRESENTATION BE STRONGLY RELIED UPON TO IMPLY THE ASYMPTOTIC PERFORMANCE?

Previous COMRL endeavors (Li et al., 2020; Yuan & Lu, 2022; Gao et al., 2024) mostly apply t-SNE on the learned task representation to show the differentiation of the tasks. They hold the belief that better performance corresponds to better differentiation of the tasks. However, based on our analysis, the visualization result only depends on the task representation at convergence and ignores the task representation shift during the whole optimization process. According to the illustration shown in Figure 5, less desirable differentiation results can also lead to better performance. Hence, the visualization results may represent the true task distribution but cannot sufficiently imply the final performance.

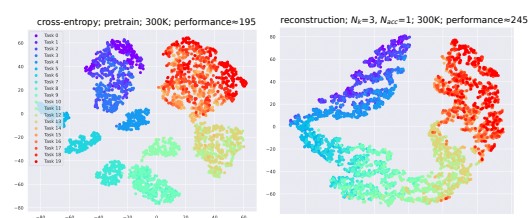

Figure 5: **The 2D projection of the learned task representation space in Ant-Dir.** Points are uniformly sampled from the evaluation datasets. Tasks of given goals from 0 to 6 are mapped to rainbow colors, ranging from purple to red.

## 7 CONCLUSION & LIMITATION

Finding that the performance improvement guarantee of the COMRL training framework remains under-explored, we constructively provide a theoretical framework to link the training scheme in COMRL with the general RL objective of maximizing the expected return. After scrutinizing this framework, we further find it ignores the variation of the task representation, which may impair performance improvement. Based on this finding, we propose a new issue called task representation shift, refine the condition for monotonicity, and prove that monotonic performance improvements can be achieved with appropriate context encoder updates. We prospect that deeply exploring the role of task representation shift can make a profound difference in the COMRL setting. Naturally, one limitation is that there may exist more advanced algorithms to control the task representation shift. Additionally, our work focuses on the representation part for monotonic performance improvement, leaving the policy learning part alone. Hence, one direction that merits further research is to design a policy learning algorithm to achieve better policy improvements conditioning on the optimal task representation distribution. We leave these interesting problems to future work.

### ACKNOWLEDGMENTS

The work is supported by the National Key Research and Development Program of China (No. 2020YFA0711402), by the Young Scientists Fund of National Natural Science Foundation of China (No. 62406295), and by the "Pioneer" and "Leading Goose" R&D Program of Zhejiang (2024SSYS0007).

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

# 8 APPENDIX

## 8.1 USEFUL LEMMAS

**Lemma 8.1** (Return bound.(Zhang et al., 2023a)). *Let $R_{max}$ denote the bound of the reward function, $\epsilon_\pi$ denote $\max_s D_{TV}(\pi_1||\pi_2)$ and $\epsilon_{M_1}^{M_2}$ denote $\mathbb{E}_{(s,a)\sim d_{M_1}^{\pi_1}}[D_{TV}(p_{M_1}||p_{M_2})]$. For two arbitrary policies $\pi_1, \pi_2 \in \Pi$, the expected return under two arbitrary models $M_1, M_2 \in \mathcal{M}$ can be bounded as,*

$$|V_{M_2}^{\pi_2} - V_{M_1}^{\pi_1}| \le 2R_{max}(\frac{\epsilon_\pi}{(1-\gamma)^2} + \frac{\gamma}{(1-\gamma)^2}\epsilon_{M_1}^{M_2}) \tag{13}$$

*Proof:*

$$|V_{M_2}^{\pi_2} - V_{M_1}^{\pi_1}| = |\sum_{t=0}^{\infty} \gamma^t \sum_{s,a}(p_{t,M_2}^{\pi_2}(s,a) - p_{t,M_1}^{\pi_1}(s,a))r(s,a)|$$

$$\le R_{max}\sum_{t=0}^{\infty} \gamma^t \sum_{s,a}|p_{t,M_2}^{\pi_2}(s,a) - p_{t,M_1}^{\pi_1}(s,a)| \tag{14}$$

$$= 2R_{max}\sum_{t=0}^{\infty}\gamma^t D_{TV}(p_{t,M_1}^{\pi_1}(s,a)||p_{t,M_2}^{\pi_2}(s,a))$$

According to Theorem 2 (Return Bound) in (Zhang et al., 2023a),

$$\sum_{t=0}^{\infty}\gamma^t D_{TV}(p_{t,M_1}^{\pi_1}(s,a)||p_{t,M_2}^{\pi_2}(s,a)) \le (\frac{\epsilon_\pi}{(1-\gamma)^2} + \frac{\gamma}{(1-\gamma)^2}\epsilon_{M_1}^{M_2}) \tag{15}$$

.

We bring this result back and can draw the conclusion.

$\square$

**Lemma 8.2** (Inequility for $L_1$ deviation of the empirical distribution.(Weissman et al., 2003)). *Let $P$ be a probability distribution on the set $\mathcal{A} = \{1, ..., a\}$. For a sequence of samples $x_1, ..., x_m \sim P$, let $\hat{P}$ be the empirical probability distribution on $\mathcal{A}$ defined by $\hat{P}(j) = \frac{1}{m}\sum_{i=1}^{m}\mathbb{I}(x_i = j)$. The $L_1$-deviation of the true distribution $P$ and the empirical distribution $\hat{P}$ over $\hat{A}$ from $m$ independent identically samples is bounded by,*

$$Pr(|P - \hat{P}| \ge \epsilon) \le (2^{|\mathcal{A}|} - 2)\exp(-m\epsilon^2/2) \tag{16}$$

*Proof:*

According to the inequality for $L_1$ deviation of the empirical distribution (Weissman et al., 2003), they conclude that $Pr(|P - \hat{P}|_1 \ge \epsilon) \le (2^{|\mathcal{A}|} - 2)\exp(-m\phi(\pi_P)\epsilon^2/4)$, where $\phi(p) = \frac{1}{1-2p}\log\frac{1-p}{p}$ and $\pi_P = \max_{A\in\mathcal{A}}\min(P(A), 1 - P(A))$. The paper (Weissman et al., 2003) point out that $0 \le \pi_P \le 1/2, \forall P$. Based on this, we can derive the function $\phi(p)$ and find that $\phi(p)$ is monotonically decreasing on $(0, 1/2)$. Therefore, we can get $\phi(p) \ge \phi(1/2) = 2$. Then, we bring this result back and can conclude that $Pr(|P - \hat{P}|_1 \ge \epsilon) \le (2^{|\mathcal{A}|} - 2)\exp(-m\epsilon^2/2)$.

$\square$

## 8.2 MISSING PROOFS

**Theorem 8.3** (Return bound in COMRL). *Assume the reward function is upper-bounded by $R_{max}$. For an arbitrary policy parameter $\theta$, when meeting Assumption 4.1, the return bound in COMRL can be formalized as:*

$$|J^*(\theta) - J(\theta)| \le \frac{2R_{max}L_z}{(1-\gamma)^2}\mathbb{E}_{m,x}(|Z(\cdot|x;\phi) - Z(\cdot|x;\phi^{mutual})| + |Z(\cdot|x;\phi^{mutual}) - Z(\cdot|x;\phi^*)|) \tag{17}$$

*Proof:*

$|J(\theta) - J^*(\theta)|$

$$= |\mathbb{E}_{m,x}[\mathbb{E}_{(s,a)\sim d_{\pi(\cdot|s,Z(\cdot|x;\phi);\theta)}}[R_m(s,a)]] - \mathbb{E}_{m,x}[\mathbb{E}_{(s,a)\sim d_{\pi(\cdot|s,Z(\cdot|x;\phi^*);\theta)}}[R_m(s,a)]]| \tag{18}$$

$$= |\mathbb{E}_{m,x}[\mathbb{E}_{(s,a)\sim d_{\pi(\cdot|s,Z(\cdot|x;\phi);\theta)}}[R_m(s,a)] - \mathbb{E}_{(s,a)\sim d_{\pi(\cdot|s,Z(\cdot|x;\phi^*);\theta)}}[R_m(s,a)]]| \tag{19}$$

$$= |\mathbb{E}_{m,x}[\mathbb{E}_{(s,a)\sim d_{\pi(\cdot|s,Z(\cdot|x;\phi);\theta)}}[R_m(s,a)] - \mathbb{E}_{(s,a)\sim d_{\pi(\cdot|s,Z(\cdot|x;\phi^{mutual});\theta)}}[R_m(s,a)]$$
$$+ \mathbb{E}_{(s,a)\sim d_{\pi(\cdot|s,Z(\cdot|x;\phi^{mutual});\theta)}}[R_m(s,a)] - \mathbb{E}_{(s,a)\sim d_{\pi(\cdot|s,Z(\cdot|x;\phi^*);\theta)}}[R_m(s,a)]]| \tag{20}$$

$$\leq \mathbb{E}_{m,x}[|\mathbb{E}_{(s,a)\sim d_{\pi(\cdot|s,Z(\cdot|x;\phi);\theta)}}[R_m(s,a)] - \mathbb{E}_{(s,a)\sim d_{\pi(\cdot|s,Z(\cdot|x;\phi^{mutual});\theta)}}[R_m(s,a)]|$$
$$+ |\mathbb{E}_{(s,a)\sim d_{\pi(\cdot|s,Z(\cdot|x;\phi^{mutual});\theta)}}[R_m(s,a)] - \mathbb{E}_{(s,a)\sim d_{\pi(\cdot|s,Z(\cdot|x;\phi^*);\theta)}}[R_m(s,a)]|] \tag{21}$$

$$= \mathbb{E}_{m,x}[|V_m^{\pi(\cdot|s,Z(\cdot|x;\phi);\theta)} - V_m^{\pi(\cdot|s,Z(\cdot|x;\phi^{mutual});\theta)}| + |V_m^{\pi(\cdot|s,Z(\cdot|x;\phi^{mutual});\theta)} - V_m^{\pi(\cdot|s,Z(\cdot|x;\phi^*);\theta)}|] \tag{22}$$

$$\leq \mathbb{E}_{m,x}[\frac{2R_{max}}{(1-\gamma)^2}\max_s D_{TV}(\pi(\cdot|s,Z(\cdot|x;\phi);\theta)||\pi(\cdot|s,Z(\cdot|x;\phi^{mutual});\theta))$$
$$+ \frac{2R_{max}}{(1-\gamma)^2}\max_s D_{TV}(\pi(\cdot|s,Z(\cdot|x;\phi^{mutual});\theta)||\pi(\cdot|s,Z(\cdot|x;\phi^*);\theta))] \tag{23}$$

$$\leq \frac{2R_{max}L_z}{(1-\gamma)^2}\mathbb{E}_{m,x}(|Z(\cdot|x;\phi) - Z(\cdot|x;\phi^{mutual})| + |Z(\cdot|x;\phi^{mutual}) - Z(\cdot|x;\phi^*)|) \tag{24}$$

Eq. (23) is the result of directly applying Lemma 8.1 and Eq. (24) is the result of directly applying Assumption 4.1. □

**Corollary 8.4** (Monotonic performance improvement condition for previous COMRL works). *When meeting Assumption 4.1, the condition for monotonic performance improvement of previous COMRL works is:*

$$\epsilon_{12}^* \triangleq J^*(\theta_2) - J^*(\theta_1) \geq \frac{4R_{max}L_z}{(1-\gamma)^2}\mathbb{E}_{m,x}(|Z(\cdot|x;\phi) - Z(\cdot|x;\phi^*)|) \tag{25}$$

*Proof:*

$$J(\theta_2) - J(\theta_1) = J(\theta_2) - J^*(\theta_2) + J^*(\theta_2) - J^*(\theta_1) + J^*(\theta_1) - J(\theta_1) \tag{26}$$

Denote $\frac{2R_{max}}{(1-\gamma)^2}$ as $\kappa$. Taking Lemma 8.1 and Assumption 4.1 into the above formulation, we can get:

$$J(\theta_2) - J^*(\theta_2) + J^*(\theta_2) - J^*(\theta_1) + J^*(\theta_1) - J(\theta_1) \tag{27}$$

$$\geq \mathbb{E}_{m,x}[-\kappa \max_s D_{TV}(\pi(\cdot|s,Z(\cdot|x;\phi);\theta_2)||\pi(\cdot|s,Z(\cdot|x;\phi^*);\theta_2))] + \epsilon_{12}^* \tag{28}$$

$$- \kappa \max_s D_{TV}(\pi(\cdot|s,Z(\cdot|x;\phi);\theta_1)||\pi(\cdot|s,Z(\cdot|x;\phi^*);\theta_1))] \tag{29}$$

$$\geq \mathbb{E}_{m,x}[-2\kappa L_z|Z(\cdot|x;\phi) - Z(\cdot|x;\phi^*)| + \epsilon_{12}^*] \tag{30}$$

To get the monotonic performance improvement, we need Eq. (30) to be larger than 0. Hence, we can get:

$$\epsilon_{12}^* - \frac{4R_{max}L_z}{(1-\gamma)^2}\mathbb{E}_{m,x}[|Z(\cdot|x;\phi) - Z(\cdot|x;\phi^*)|] \geq 0 \tag{31}$$

□

**Theorem 8.5** (Lower bound of performance difference in COMRL). *Assume the reward function is upper-bounded by $R_{max}$. When meeting Assumption 4.1, the lower bound of the performance difference in COMRL can be formalized as:*

$$J^2(\theta_2) - J^1(\theta_1) \geq \epsilon_{12}^* - \frac{2R_{max}L_z}{(1-\gamma)^2}\mathbb{E}_{m,x}[2|Z(\cdot|x;\phi_2) - Z(\cdot|x;\phi^*)| + |Z(\cdot|x;\phi_2) - Z(\cdot|x;\phi_1)|] \tag{32}$$

*Proof:* We can introduce the following decomposition

$$J^2(\theta_2) - J^1(\theta_1) \tag{33}$$

$$= \mathbb{E}_{m,x}\mathbb{E}_{(s,a)\sim d_{\pi(\cdot|s,Z(\cdot|x;\phi_2);\theta_2)}}[R_m(s,a)] - \mathbb{E}_{m,x}\mathbb{E}_{(s,a)\sim d_{\pi(\cdot|s,Z(\cdot|x;\phi_1);\theta_1)}}[R_m(s,a)] \tag{34}$$

$$= \mathbb{E}_{m,x}[\mathbb{E}_{(s,a)\sim d_{\pi(\cdot|s,Z(\cdot|x;\phi_2);\theta_2)}}[R_m(s,a)] - \mathbb{E}_{(s,a)\sim d_{\pi(\cdot|s,Z(\cdot|x;\phi_1);\theta_1)}}[R_m(s,a)]] \tag{35}$$

Denote $\mathbb{E}_{(s,a)\sim d_{\pi(\cdot|s,Z_i(\cdot|x);\theta_j)}}[R_m(s,a)]]$ as $G_{Z_i}^{\theta_j}$, then we can get the following derivation. Here, we only consider the value within the brackets.

$$= \mathbb{E}_{(s,a)\sim d_{\pi(\cdot|s,Z(\cdot|x;\phi_2);\theta_2)}}[R_m(s,a)] - \mathbb{E}_{(s,a)\sim d_{\pi(\cdot|s,Z(\cdot|x;\phi_1);\theta_1)}}[R_m(s,a)] \tag{36}$$

$$= G_{Z_2}^{\theta_2} - G_{Z_1}^{\theta_1} \tag{37}$$

$$= G_{Z_2}^{\theta_2} - G_{Z^*}^{\theta_2} + G_{Z^*}^{\theta_2} - G_{Z^*}^{\theta_1} + G_{Z^*}^{\theta_1} - G_{Z_2}^{\theta_1} + G_{Z_2}^{\theta_1} - G_{Z_1}^{\theta_1} \tag{38}$$

Denote $\frac{2R_{max}}{(1-\gamma)^2}$ as $\kappa$. Taking Lemma 8.1 and Assumption 4.1 into the above formulation, we can get:

$$G_{Z_2}^{\theta_2} - G_{Z^*}^{\theta_2} + G_{Z^*}^{\theta_2} - G_{Z^*}^{\theta_1} + G_{Z^*}^{\theta_1} - G_{Z_2}^{\theta_1} + G_{Z_2}^{\theta_1} - G_{Z_1}^{\theta_1} \tag{39}$$

$$\geq -\kappa \max_s D_{TV}(\pi(\cdot|s,Z(\cdot|x;\phi_2);\theta_2)||\pi(\cdot|s,Z(\cdot|x;\phi^*);\theta_2)) + \epsilon_{12}^*$$

$$- \kappa \max_s D_{TV}(\pi(\cdot|s,Z(\cdot|x;\phi^*);\theta_1)||\pi(\cdot|s,Z(\cdot|x;\phi_2);\theta_1))$$

$$- \kappa \max_s D_{TV}(\pi(\cdot|s,Z(\cdot|x;\phi_2);\theta_1)||\pi(\cdot|s,Z(\cdot|x;\phi_1);\theta_1)) \tag{40}$$

$$\geq -\kappa L_z|Z(\cdot|x;\phi_2) - Z(\cdot|x;\phi^*)| + \epsilon_{12}^*$$

$$- \kappa L_z|Z(\cdot|x;\phi_2) - Z(\cdot|x;\phi^*)| - \kappa L_z|Z(\cdot|x;\phi_2) - Z(\cdot|x;\phi_1)| \tag{41}$$

Simplifying Eq. (41) we can get:

$$J^2(\theta_2) - J^1(\theta_1) \geq \epsilon_{12}^* - \frac{2R_{max}L_z}{(1-\gamma)^2}\mathbb{E}_{m,x}[2|Z(\cdot|x;\phi_2) - Z(\cdot|x;\phi^*)| + |Z(\cdot|x;\phi_2) - Z(\cdot|x;\phi_1)|] \tag{42}$$

$\square$

**Theorem 8.6** (Monotonic performance improvement guarantee on training process). *Denote $\kappa$ as $\frac{(1-\gamma)^2}{4R_{max}L_z}\epsilon_{12}^* - \frac{1}{2}\beta$ and $|Z|$ as the cardinality of the task representation space. Given that the context encoder has already been trained by maximizing $I(Z;M)$ to some extent. When meeting Assumption 4.1, 4.7, 4.8 and 4.9, with a probability greater than $1 - \xi$, we can get the monotonic performance improvement guarantee by updating the context encoder via maximizing $I(Z;M)$ from at least extra $k$ samples, where:*

$$k = \frac{1}{\kappa^2}(\sqrt{2\log\frac{2^{|Z|} - 2}{\xi}} + \sqrt{\alpha})^2 \tag{43}$$

*Here, $\xi \in [0,1]$ is a constant.*

*Proof:*

Let $Z_1$ have already been trained by maximizing $I(Z;M)$ to some extent. Given that to get $Z_2$, we need to train $Z_1$ by extra $k$ samples from the training dataset by maximizing $I(Z;M)$ to get the monotonic performance improvement guarantee.

We begin with the following inequality:

$$|Z(\cdot|x;\phi_2) - Z(\cdot|x;\phi^*)| \leq |Z(\cdot|x;\phi_2) - Z(\cdot|x;\tilde{\phi}_2)| + |Z(\cdot|x;\tilde{\phi}_2) - Z(\cdot|x;\phi^*)| \tag{44}$$

where $Z(\cdot|x;\tilde{\phi}_2)$ denotes the context encoder updated by fitting the empirical distribution on $k$ i.i.d samples from $Z(\cdot|x;\phi^*)$ based on $Z(\cdot|x;\phi_1)$.

According to Assumption 4.9, we have:

$$|Z(\cdot|x;\phi_2) - Z(\cdot|x;\tilde{\phi}_2)| \leq \sqrt{\frac{\alpha}{k}} \tag{45}$$

Now, we move our focus on solving $|Z(\cdot|x;\tilde{\phi}_2) - Z(\cdot|x;\phi^*)|$.

By applying Lemma 8.2, the $L_1$ deviation of the empirical distribution $Z(\cdot|x;\tilde{\phi}_2)$ and true $Z(\cdot|x;\phi^*)$ over $|Z|$ is bounded by:

$$Pr(|Z(\cdot|x;\tilde{\phi}_2) - Z(\cdot|x;\phi^*)| \geq \epsilon) \leq (2^{|Z|} - 2)\exp{(-k\epsilon^2/2)} \tag{46}$$

$$Pr(|Z(\cdot|x;\tilde{\phi}_2) - Z(\cdot|x;\phi^*)| <= \epsilon) \geq 1 - (2^{|Z|} - 2)\exp{(-k\epsilon^2/2)} \tag{47}$$

Then for a fixed $x$, with probability greater than $1 - \xi$, we have:

$$|Z(\cdot|x; \tilde{\phi}_2) - Z(\cdot|x; \phi^*)| \leq \sqrt{\frac{2}{k} \log \frac{2^{|Z|} - 2}{\xi}} \tag{48}$$

Hence, we can get:

$$|Z(\cdot|x; \phi_2) - Z(\cdot|x; \phi^*)| \leq \sqrt{\frac{2}{k} \log \frac{2^{|Z|} - 2}{\xi}} + \sqrt{\frac{\alpha}{k}} \tag{49}$$

Let $\epsilon = \frac{(1-\gamma)^2}{4R_{max}L_z}\epsilon_{12}^* - \frac{1}{2}\beta$, and recall that $\kappa$ denotes $\frac{(1-\gamma)^2}{4R_{max}L_z}\epsilon_{12}^* - \frac{1}{2}\beta$, then we can get the $k$ as:

$$\sqrt{\frac{2}{k} \log \frac{2^{|Z|} - 2}{\xi}} + \sqrt{\frac{\alpha}{k}} \leq \frac{(1-\gamma)^2}{4R_{max}L_z}\epsilon_{12}^* - \frac{1}{2}\beta \tag{50}$$

$$k \geq \frac{1}{\kappa^2}(\sqrt{2 \log \frac{2^{|Z|} - 2}{\xi}} + \sqrt{\alpha})^2 \tag{51}$$

Since we need the least number of samples that update the context encoder, we take $k = \frac{1}{\kappa^2}(\sqrt{2 \log \frac{2^{|Z|}-2}{\xi}} + \sqrt{\alpha})^2$. $\qquad\square$

**Corollary 8.7** (Monotonic performance improvement condition for pre-training scheme)**.** *Denote $Z(\cdot|x; \phi^{pretrain})$ as the task representation distribution after pre-training. When meeting Assumption 4.1, the monotonic performance improvement condition for pre-training scheme is:*

$$\epsilon_{12}^* - \frac{4R_{max}L_z}{(1-\gamma)^2}\mathbb{E}_{m,x}[|Z(\cdot|x; \phi^{pretrain}) - Z(\cdot|x; \phi^*)|] \geq 0 \tag{52}$$

*Proof:* Denote $J^{pretrain}(\theta_1)$ as the expected return of the policy $\pi(\cdot|s, Z(\cdot|x; \phi^{pretrain}); \theta_1)$ before update of the policy. Similarly, denote $J^{pretrain}(\theta_2)$ as the expected return of the policy $\pi(\cdot|s, Z(\cdot|x; \phi^{pretrain}); \theta_2)$ after update of the policy.

$$J^{pretrain}(\theta_2) - J^{pretrain}(\theta_1) \tag{53}$$

$$= J^{pretrain}(\theta_2) - J^*(\theta_2) + J^*(\theta_2) - J^*(\theta_1) + J^*(\theta_1) - J^{pretrain}(\theta_1) \tag{54}$$

Denote $\frac{2R_{max}}{(1-\gamma)^2}$ as $\kappa$. Taking Lemma 8.1 and Assumption 4.1 into the above formulation, we can get:

$$J^{pretrain}(\theta_2) - J^*(\theta_2) + J^*(\theta_2) - J^*(\theta_1) + J^*(\theta_1) - J^{pretrain}(\theta_1) \tag{55}$$

$$\geq \mathbb{E}_{m,x}[-\kappa \max_s D_{TV}(\pi(\cdot|s, Z(\cdot|x; \phi^{pretrain}); \theta_2)||\pi(\cdot|s, Z(\cdot|x; \phi^*); \theta_2))] + \epsilon_{12}^* \tag{56}$$

$$- \kappa \max_s D_{TV}(\pi(\cdot|s, Z(\cdot|x; \phi^{pretrain}); \theta_1)||\pi(\cdot|s, Z(\cdot|x; \phi^*); \theta_1))] \tag{57}$$

$$\geq \mathbb{E}_{m,x}[-2\kappa L_z|Z(\cdot|x; \phi^{pretrain}) - Z(\cdot|x; \phi^*)| + \epsilon_{12}^*] \tag{58}$$

To get the monotonic performance improvement, we need Eq. (58) to be larger than 0. Hence, we can get:

$$\epsilon_{12}^* - \frac{4R_{max}L_z}{(1-\gamma)^2}\mathbb{E}_{m,x}[|Z(\cdot|x; \phi^{pretrain}) - Z(\cdot|x; \phi^*)|] \geq 0 \tag{59}$$

$\qquad\square$

## 8.3 JUSTIFICATION OF ASSUMPTIONS

**Assumption 4.7:**

As our aim is to rein in the task representation shift, setting a threshold to bound the task representation shift is natural. Nevertheless, how to set this threshold smartly or automatically adjust this threshold would need further research and we leave this to future work. The task representation shift less

than the policy improvement $\epsilon_{12}^*$ with a certain coefficient is to help us ensure $\kappa$ in Theorem 4.10 is larger than 0. Since we can update the policy consistently (e.g. $k$ determines when to update the context encoder, but the policy is updated normally), $\epsilon_{12}^*$ can accumulate gradually. Therefore, this assumption is reasonable.

**Assumption 4.8:**

Since the task representation is obtained through sampling, whether during the guidance of the downstream policy or the training of the context encoder by maximizing $I(Z; M)$, it is reasonable to assume that the space of the task representation is discrete and limited. We think this assumption is general as it can cover various cases, e.g. sampling, generated from the deterministic network, or discretization.

**Assumption 4.9:**

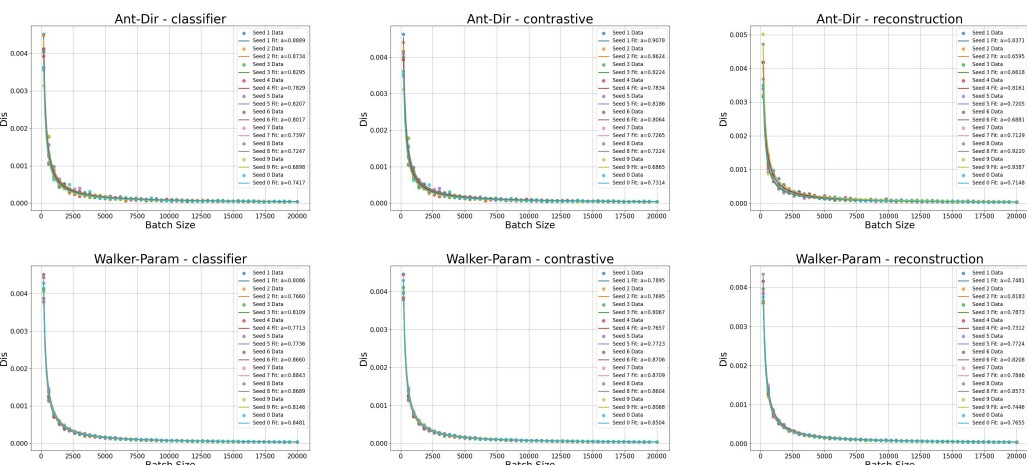

Figure 6: The numerical experiments on Ant-Dir and Walker-Param.

To justify this assumption in practice, we design a numerical experiment.

Firstly, notice that the sub-optimality gap between $Z(\cdot|x; \phi^{mutual})$ and $Z(\cdot|x; \phi^*)$ is a constant. We can introduce a notation $Z(\cdot|x; \hat{\phi}_2)$ that denotes the context encoder updated by fitting the empirical distribution on $b$ i.i.d samples from $Z(\cdot|x; \phi^{mutual})$ based on $Z(\cdot|x; \phi_1)$. Similar to Eq. (48), we can get the upper bound of the discrepancy between $Z(\cdot|x; \hat{\phi}_2)$ and $Z(\cdot|x; \tilde{\phi}_2)$:

$$|Z(\cdot|x; \hat{\phi}_2) - Z(\cdot|x; \tilde{\phi}_2)| \tag{60}$$

$$\leq |Z(\cdot|x; \hat{\phi}_2) - Z(\cdot|x; \phi^{mutual})| + |Z(\cdot|x; \phi^{mutual}) - Z(\cdot|x; \phi^*)| + |Z(\cdot|x; \phi^*) - Z(\cdot|x; \tilde{\phi}_2)| \tag{61}$$

$$= \sqrt{\frac{constant1}{b}} + \underbrace{|Z(\cdot|x; \phi^{mutual}) - Z(\cdot|x; \phi^*)|}_{\text{sub-optimality gap}} + \sqrt{\frac{constant2}{b}} \tag{62}$$

$$= \sqrt{\frac{constant3}{b}} + constant4 \tag{63}$$

Secondly, we use the pre-trained context encoder to be $Z(\cdot|x; \phi^{mutual})$, use the initialized context encoder to be $Z(\cdot|x; \phi_1)$ and simulate two update processes: 1) use the specific objective w.r.t maximizing $I(Z; M)$ to update $Z(\cdot|x; \phi_1)$ to get $Z(\cdot|x; \phi_2)$; 2) use task representations randomly sampled from the pre-trained context encoder to update $Z(\cdot|x; \phi_1)$ to get $Z(\cdot|x; \hat{\phi}_2)$. We set different numbers of samples to complete these two update processes.

To express the discrepancy, we compute $nn.MSE$ loss of the task representation randomly sampled from $Z(\cdot|x; \phi_2)$ and $Z(\cdot|x; \hat{\phi}_2)$. As shown in Figure 6, it does fit an inversely proportional trend as the training size increases. Therefore, we can practically denote the discrepancy between $Z(\cdot|x; \phi_2)$

and $Z(\cdot|x; \hat{\phi}_2)$ as:

$$|Z(\cdot|x; \phi_2) - Z(\cdot|x; \hat{\phi}_2)| \leq \sqrt{\frac{\text{constant5}}{b}} \qquad (64)$$

By combining the above formulations, we can get:

$$|Z(\cdot|x; \phi_2) - Z(\cdot|x; \tilde{\phi}_2)| \qquad (65)$$

$$\leq |Z(\cdot|x; \phi_2) - Z(\cdot|x; \hat{\phi}_2)| + |Z(\cdot|x; \hat{\phi}_2) - Z(\cdot|x; \tilde{\phi}_2)| \qquad (66)$$

$$= \sqrt{\frac{\text{constant6}}{b}} + \text{constant4} \qquad (67)$$

Since the training sample is finite, the discrepancy would not converge to 0. Hence, with an appropriate $\alpha$, Assumption 4.9 can be grounded in practice.

Furthermore, even if we assume the discrepancy would converge to 0 and take the constant into Assumption 4.9, it only needs to modify the formulation of $\kappa$ in Theorem 4.10.

*Proof:* If we take the constant $c$ into Assumption 4.9, then it becomes:

$$|Z(\cdot|x; \phi_2) - Z(\cdot|x; \tilde{\phi}_2)| \leq |Z(\cdot|x; \phi_2) - Z(\cdot|x; \hat{\phi}_2)| + |Z(\cdot|x; \hat{\phi}_2) - Z(\cdot|x; \tilde{\phi}_2)| = \sqrt{\frac{\alpha}{b}} + c \qquad (68)$$

Therefore, we need to modify Eq. (49) as:

$$|Z(\cdot|x; \phi_2) - Z(\cdot|x; \phi^*)| \leq \sqrt{\frac{2}{k} \log \frac{2^{|Z|} - 2}{\xi}} + \sqrt{\frac{\alpha}{k}} + c \qquad (69)$$

Let $\epsilon = \frac{(1-\gamma)^2}{4R_{max}L_z} \epsilon_{12}^* - \frac{1}{2}\beta$, and update the notation of $\kappa$ as $\frac{(1-\gamma)^2}{4R_{max}L_z} \epsilon_{12}^* - \frac{1}{2}\beta - c$, then we can get the $k$ as:

$$\sqrt{\frac{2}{k} \log \frac{2^{|Z|} - 2}{\xi}} + \sqrt{\frac{\alpha}{k}} + c \leq \frac{(1-\gamma)^2}{4R_{max}L_z} \epsilon_{12}^* - \frac{1}{2}\beta \qquad (70)$$

$$k \geq \frac{1}{\kappa^2} \left(\sqrt{2 \log \frac{2^{|Z|} - 2}{\xi}} + \sqrt{\alpha}\right)^2 \qquad (71)$$

Since we need the least number of samples that update the context encoder, we take $k = \frac{1}{\kappa^2}\left(\sqrt{2 \log \frac{2^{|Z|}-2}{\xi}} + \sqrt{\alpha}\right)^2$. $\qquad \square$

To further enhance the theoretical rigor, we can even set the bound of Assumption 4.9 as a constant since the bound between $Z(\cdot|x; \phi_2)$ and $Z(\cdot|x; \tilde{\phi}_2)$ will be reduced consistently with larger data sizes. Setting this as a constant only needs the accumulation of $\epsilon_{12}^*$ to become larger, which is also acceptable under our theoretical framework. For specific algorithms, e.g. contrastive, reconstruction,..., if there exist theoretical guarantees for the convergence bound, it is nice to derive a more precise calculation method for $k$.

*Proof:* If we set the bound in Assumption 4.9 to be a constant $\alpha$, then Eq. (49) becomes:

$$|Z(\cdot|x; \phi_2) - Z(\cdot|x; \phi^*)| \leq \sqrt{\frac{2}{k} \log \frac{2^{|Z|} - 2}{\xi}} + \sqrt{\alpha} \qquad (72)$$

Let $\epsilon = \frac{(1-\gamma)^2}{4R_{max}L_z} \epsilon_{12}^* - \frac{1}{2}\beta$, and update the notation of $\kappa$ as $\frac{(1-\gamma)^2}{4R_{max}L_z} \epsilon_{12}^* - \frac{1}{2}\beta - \sqrt{\alpha}$, then we can get the $k$ as:

$$\sqrt{\frac{2}{k} \log \frac{2^{|Z|} - 2}{\xi}} + \sqrt{\alpha} \leq \frac{(1-\gamma)^2}{4R_{max}L_z} \epsilon_{12}^* - \frac{1}{2}\beta \qquad (73)$$

$$k \geq \frac{2}{\kappa^2} \log \frac{2^{|Z|} - 2}{\xi} \qquad (74)$$

Since we need the least number of samples that update the context encoder, we take $k = \frac{2}{\kappa^2} \log \frac{2^{|Z|}-2}{\xi}$.

$\square$

## 8.4 Implementation Details

In this section, we report the details of all the objectives in our main paper.

**Contrastive-based (Li et al., 2020).** The contrastive-based algorithm uses the task representation to compute distance metric learning loss. We adopt the open-source code of FOCAL[3].

**Reconstruction-based (Li et al., 2024).** The reconstruction-based algorithm passes the state, action, and task representation through the decoder to reconstruct the next state and the reward signal. We use the open-source code of UNICORN [4] and adopt UNICORN-0 as the reconstruction-based objective training framework.

**Cross-entropy-based.** The cross-entropy-based algorithm is a straightforward discrete approximation to $I(Z; M)$. Specifically, after extracting from the context encoder, the task representation would be passed through a fully connected layer to get the probabilities for each task. Then, the context encoder is trained by the supervision of the task label and back-propagation of the cross-entropy loss.

According to Theorem 3.1, the previous methods are optimizing the approximate bounds of $I(Z; M)$. To better approximate $I(Z; M)$, we choose to face this term directly. Notice that $I(Z; M) = H(M) - H(M|Z)$. For the first part, it is a constant w.r.t the variable $Z$. Hence, it can be ignored. For the second part, we have:

$$H(M|Z) = -\mathbb{E}_z[\sum_m p(m|z) \log p(m|z)] \tag{75}$$

The ideal condition is task representation $Z$ can uniquely identify the task $M$. Hence, minimizing $H(M|Z)$ is equivalent to maximizing $\mathbb{E}_z[\log p(m|z)]$, as we need $p(m|z)$ to approach 1. As maximizing $\log p(m|z)$ can be instantiated as the cross-entropy-loss, we claim that cross-entropy-loss is a direct approximation towards optimizing $I(Z; M)$.

Built upon the code base of FOCAL, cross-entropy-based only replaces the distance metric learning loss with cross-entropy loss.

To maintain fairness, we make sure all benchmark-irrelevant parameters are consistent, as shown in Table 1.

Table 1: Benchmark-irrelevant parameters setting in the training process.

| | |
|---|---|
| training tasks | 20 |
| testing tasks | 20 |
| task training batch size | 16 |
| rl batch size | 256 |
| context size | 1 trajectory |
| actor-network size | [256, 256] |
| critic-network size | [256, 256] |
| task encoder network size | [64, 64] |
| learning rate | 3e-4 |

For each environment, we make the benchmark-relevant parameters the same for each algorithm, as shown in Table 2.

Table 3 reports the performance in our experimental setting. The results are averaged by 8 random seeds with each seed averaged by the last 5 evaluation performances. Furthermore, to demonstrate whether the performance improvement w.r.t the worst case is statistically significant, we conduct a paired $t$-test for the experimental results. The statistically significant results are highlighted in red while others are colored in blue.

---

[3]https://github.com/FOCAL-ICLR/FOCAL-ICLR/
[4]https://github.com/betray12138/UNICORN.git/

Table 2: Benchmark-relevant parameters setting in the training process.

| Configurations | Reach | Ant-Dir | Button-Press | Dial-Turn | Walker-Param | Push |
|---|---|---|---|---|---|---|
| dataset size | 3e5 | 9e4 | 3e5 | 3e5 | 4.5e5 | 3e5 |
| task representation dimension | 5 | 5 | 5 | 5 | 5 | 5 |

Table 3: The performance in our experiments section. Each result is averaged by 8 random seeds.

| Benchmark | Algorithms | $N_k = 1, N_{acc} = 3$ | $N_k = 1, N_{acc} = 2$ | $N_k = 1, N_{acc} = 1$ (original) | $N_k = 2, N_{acc} = 1$ | $N_k = 3, N_{acc} = 1$ |
|---|---|---|---|---|---|---|
| Reach | Cross-entropy | $\mathbf{2856.35 \pm 315.30}$ ($p = 0.017$) | $2573.235 \pm 245.25$ ($p = 0.482$) | $2470.25 \pm 300.93$ ($p = 0.948$) | $2457.77 \pm 309.70$ | $2688.17 \pm 166.59$ ($p = 0.034$) |
| | Contrastive | $2622.02 \pm 228.00$ ($p = 0.667$) | $2787.83 \pm 148.31$ ($p = 0.043$) | $2580.39 \pm 141.73$ | $\mathbf{2802.88 \pm 163.66}$ ($p = 0.006$) | $2616.68 \pm 418.95$ ($p = 0.828$) |
| | Reconstruction | $\mathbf{2692.67 \pm 289.51}$ ($p = 0.033$) | $2650.42 \pm 162.62$ ($p = 0.044$) | $2405.53 \pm 190.28$ | $2614.61 \pm 340.11$ ($p = 0.215$) | $2533.80 \pm 393.97$ ($p = 0.409$) |
| Dial-Turn | Cross-entropy | $\mathbf{1365.08 \pm 229.77}$ ($p = 0.049$) | $1126.04 \pm 249.26$ ($p = 0.402$) | $1015.07 \pm 300.47$ | $1167.92 \pm 208.74$ ($p = 0.200$) | $1127.70 \pm 358.05$ ($p = 0.633$) |
| | Contrastive | $\mathbf{1314.06 \pm 330.98}$ ($p = 0.024$) | $1034.83 \pm 322.80$ | $1146.31 \pm 265.92$ ($p = 0.550$) | $1118.64 \pm 270.86$ ($p = 0.599$) | $1050.41 \pm 300.35$ ($p = 0.941$) |
| | Reconstruction | $1420.08 \pm 193.14$ ($p = 0.008$) | $1337.72 \pm 256.86$ ($p = 0.081$) | $1353.98 \pm 198.90$ ($p = 0.046$) | $1111.40 \pm 215.29$ | $\mathbf{1740.45 \pm 51.78}$ ($p = 0.0002$) |
| Button-Press | Cross-entropy | $1164.41 \pm 424.59$ ($p = 0.645$) | $1422.94 \pm 517.74$ ($p = 0.132$) | $1236.10 \pm 187.42$ ($p = 0.253$) | $\mathbf{1588.21 \pm 301.15}$ ($p = 0.002$) | $1048.96 \pm 327.52$ |
| | Contrastive | $\mathbf{1573.33 \pm 138.66}$ ($p = 0.0001$) | $1167.91 \pm 396.01$ ($p = 0.060$) | $1206.87 \pm 328.34$ ($p = 0.014$) | $645.23 \pm 338.07$ | $1296.29 \pm 439.07$ ($p = 0.014$) |
| | Reconstruction | $\mathbf{2947.89 \pm 271.49}$ ($p = 0.003$) | $2121.66 \pm 601.57$ | $2361.82 \pm 517.93$ ($p = 0.367$) | $2496.67 \pm 594.00$ ($p = 0.282$) | $2297.39 \pm 559.90$ ($p = 0.561$) |
| Push | Cross-entropy | $1279.85 \pm 300.24$ ($p = 0.175$) | $889.64 \pm 158.81$ | $1103.52 \pm 323.49$ ($p = 0.139$) | $\mathbf{1461.72 \pm 157.71}$ ($p = 0.0007$) | $1338.57 \pm 259.27$ ($p = 0.034$) |
| | Contrastive | $661.95 \pm 176.91$ | $941.42 \pm 192.44$ ($p = 0.277$) | $839.05 \pm 267.61$ ($p = 0.394$) | $\mathbf{1557.84 \pm 495.95}$ ($p = 0.019$) | $1172.82 \pm 387.91$ ($p = 0.039$) |
| | Reconstruction | $\mathbf{1357.04 \pm 327.87}$ ($p = 0.038$) | $1146.68 \pm 465.36$ ($p = 0.737$) | $1063.80 \pm 266.31$ | $1071.75 \pm 194.36$ ($p = 0.977$) | $1289.08 \pm 354.33$ ($p = 0.114$) |
| Walker-Param | Cross-entropy | $365.22 \pm 69.27$ ($p = 0.127$) | $337.73 \pm 121.10$ ($p = 0.629$) | $284.56 \pm 103.71$ | $\mathbf{399.87 \pm 87.20}$ ($p = 0.042$) | $371.48 \pm 96.95$ ($p = 0.093$) |
| | Contrastive | $301.70 \pm 59.35$ ($p = 0.859$) | $333.35 \pm 40.14$ ($p = 0.283$) | $299.00 \pm 35.42$ | $\mathbf{450.47 \pm 19.71}$ ($p = 0.002$) | $432.61 \pm 77.14$ ($p = 0.022$) |
| | Reconstruction | $281.50 \pm 97.94$ ($p = 0.339$) | $240.56 \pm 88.25$ ($p = 0.763$) | $232.64 \pm 78.06$ | $\mathbf{370.97 \pm 103.82}$ ($p = 0.030$) | $292.53 \pm 134.26$ ($p = 0.178$) |
| Ant-Dir | Cross-entropy | $245.01 \pm 18.20$ | $252.73 \pm 19.95$ ($p = 0.413$) | $253.83 \pm 9.56$ ($p = 0.320$) | $\mathbf{291.29 \pm 15.69}$ ($p = 0.002$) | $283.56 \pm 17.48$ ($p = 0.002$) |
| | Contrastive | $187.72 \pm 19.64$ | $211.31 \pm 20.04$ ($p = 0.138$) | $207.92 \pm 18.01$ ($p = 0.107$) | $\mathbf{268.23 \pm 19.74}$ ($p = 0.00004$) | $259.47 \pm 16.19$ ($p = 0.0001$) |
| | Reconstruction | $205.77 \pm 21.88$ | $222.21 \pm 19.71$ ($p = 0.205$) | $213.78 \pm 31.88$ ($p = 0.434$) | $244.52 \pm 16.20$ ($p = 0.003$) | $\mathbf{247.81 \pm 23.60}$ ($p = 0.001$) |
| Ant-Dir-Random | Cross-entropy | $55.04 \pm 13.56$ ($p = 0.486$) | $54.09 \pm 9.75$ ($p = 0.760$) | $52.06 \pm 12.51$ | $\mathbf{77.47 \pm 22.20}$ ($p = 0.043$) | $52.18 \pm 26.82$ ($p = 0.991$) |
| | Contrastive | $-0.13 \pm 0.40$ | $-0.16 \pm 0.38$ | $0.05 \pm 0.30$ | $0.04 \pm 0.39$ | $-0.07 \pm 0.27$ |
| | Reconstruction | $\mathbf{53.55 \pm 17.94}$ ($p = 0.029$) | $45.61 \pm 10.05$ ($p = 0.696$) | $43.34 \pm 10.90$ ($p = 0.922$) | $51.71 \pm 9.08$ ($p = 0.086$) | $42.63 \pm 17.37$ |
| Ant-Dir-Middle | Cross-entropy | $185.10 \pm 15.83$ ($p = 0.003$) | $156.89 \pm 30.80$ ($p = 0.028$) | $140.64 \pm 37.23$ ($p = 0.021$) | $152.72 \pm 46.76$ ($p = 0.049$) | $113.96 \pm 34.49$ |
| | Contrastive | $\mathbf{203.53 \pm 19.65}$ ($p = 0.003$) | $166.72 \pm 17.95$ ($p = 0.481$) | $156.66 \pm 26.05$ | $199.96 \pm 29.38$ ($p = 0.042$) | $178.09 \pm 33.00$ ($p = 0.312$) |
| | Reconstruction | $166.75 \pm 26.74$ ($p = 0.798$) | $\mathbf{193.11 \pm 18.21}$ ($p = 0.036$) | $185.11 \pm 24.86$ ($p = 0.321$) | $178.75 \pm 25.70$ ($p = 0.350$) | $160.72 \pm 46.56$ |
| Ant-Dir-Expert | Cross-entropy | $207.21 \pm 22.30$ | $218.31 \pm 21.36$ ($p = 0.288$) | $228.97 \pm 18.16$ ($p = 0.144$) | $\mathbf{261.42 \pm 19.32}$ ($p = 0.003$) | $237.93 \pm 28.75$ ($p = 0.037$) |
| | Contrastive | $219.29 \pm 16.16$ | $246.19 \pm 26.44$ ($p = 0.056$) | $261.71 \pm 21.06$ ($p = 0.008$) | $286.68 \pm 26.53$ ($p = 0.003$) | $\mathbf{317.66 \pm 23.25}$ ($p = 0.0001$) |
| | Reconstruction | $216.05 \pm 32.18$ | $219.70 \pm 18.40$ ($p = 0.803$) | $230.49 \pm 26.76$ ($p = 0.478$) | $\mathbf{281.18 \pm 7.71}$ ($p = 0.001$) | $262.63 \pm 18.95$ ($p = 0.012$) |

## 8.5 ADDITIONAL RESULTS

### 8.5.1 PERFORMANCE CONTROL

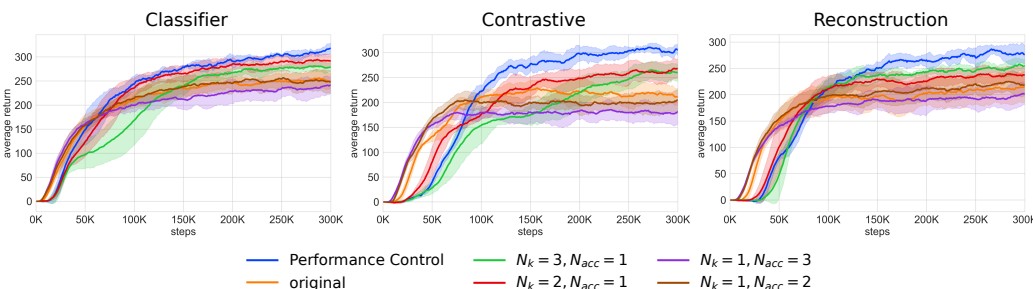

Figure 7: The experimental results of using the evaluation performance to guide the learning of the context encoder on Ant-Dir. Each result of **Performance Control** is averaged by 6 random seeds.

Here, we give a potential way to achieve better performance. As shown in Theorem 4.10, $k$ has a close connection with $\epsilon_{12}^*$. When $\epsilon_{12}^*$ is sufficient, $k$ would become less than the given batch size, then the context encoder can be updated. The straightforward way is to use the performance evaluated through online interaction with the environment as the guideline. Specifically, we use the evaluation performance of the learned policy conditioning on the pre-trained task representation to approximate $J^*(\theta)$, thereby providing a way to approximate $\epsilon_{12}^*$. To resolve performance fluctuations, we calculate $\epsilon_{12}^*$ through $\epsilon_{12}^* \leftarrow \epsilon_{12}^* + \max(J^*(\theta_2) - J^*(\theta_1), 0)$. According to Theorem 4.10, we simply set a hyper-parameter $\alpha$ to calculate $k$ as $k = \frac{\alpha}{(\epsilon_{12}^*)^2 + 1e-9}$. During this experiment, we set $\alpha = 0.07$. As for $\beta$, we simply set the context encoder to be updated once when an update is required.

As shown in Figure 7, we observe that all three algorithms exhibit improved asymptotic performance on Ant-Dir. This suggests a potential way to reduce parameter sensitivity by leveraging performance to guide the training of the context encoder.

Nevertheless, the current method is not practically feasible. Calculating online performance evaluations across 20 training tasks for each training step would require up to a month on an NVIDIA 3090 GPU. To speed up the process, we randomly select only 3 training tasks for calculating, reducing the training time to approximately 3–6 days, depending on the algorithm. **Note that the performance on learning curves is still averaged by** 20 **testing tasks.** Exploring methods to guide the training of the context encoder through more efficient performance estimation techniques could be a promising direction for future work. We also envision our work being generalized into other areas Peng et al. (2024); Zhang et al. (2024); Peng et al. (2025) that use the (context) encoder.

### 8.5.2 EVALUATION OF CSRO

Since our goal is to validate that reining in the task representation shift also makes a difference for CSRO Gao et al. (2024), we do not carefully tune its hyper-parameters. As shown in Figure 8 and Figure 9, the conclusions still hold in CSRO.

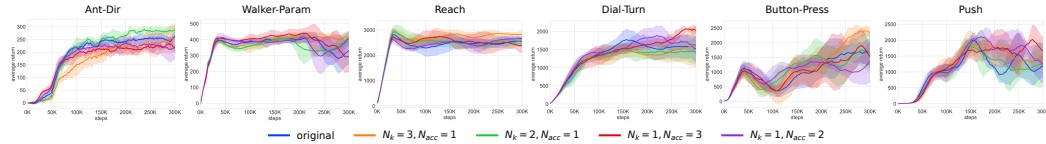

Figure 8: **Testing returns of different settings to rein in the task representation shift of CSRO.**

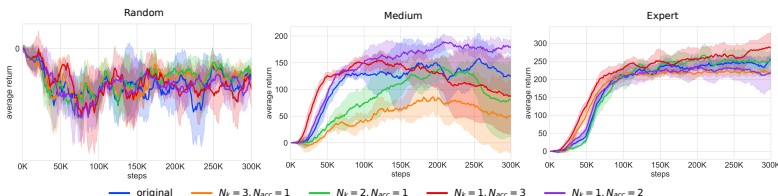

Figure 9: **Testing returns of different settings to rein in the task representation shift on different data qualities in Ant-Dir of CSRO.**

