# OpenReview forum: "Scrutinize What We Ignore: Reining In Task Representation Shift Of Context-Based Offline Meta Reinforcement Learning"
_ICLR.cc/2025/Conference — ICLR 2025 Poster_

### Official Review · Reviewer_jUhN · 2024-11-01

**Soundness:** 3
**Presentation:** 3
**Contribution:** 3
**Rating:** 6
**Confidence:** 4

**Summary:**

The paper focuses on task representation shift in context encoder updates when the context encoder is updated in offline meta reinforcement learning (OMRL). In particular, the paper analyzes the relationship between performance improvement in policy updates and the amount of data used for context encoder updates.

The paper claims two contributions:

(1) Identifying task representation shift as a major issue in context based offline meta reinforcement learning. The task representation shift makes it harder to satisfy the condition necessary for monotonic performance improvements. For the "major issue" claim, the paper claims to provide empirical evidence.

(2) Proof of monotonic performance improvement taking into account task representation shift and number of samples used to improve the context encoder. This information can be used in practice, for example, such that an algorithm updates the context encoder only when the improvement due to policy updates is sufficiently large such that the number of samples used for the context encoder updates is sufficient.

**Strengths:**

According to my knowledge the theoretical analysis of task representation shift in context based offline meta reinforcement learning is original and may be of interest to researchers working in this domain and potentially others where context encoders are used.

For claim (1), the theoretical analysis is a solid contribution in case the authors can clarify the questions/comments further below. The insight that task representation shift influences the requirements for monotonic performance improvements is valuable.

For claim (2), mathematical proofs under the made assumptions appear correct.

The paper is overall well written and understandable.

**Weaknesses:**

For claim (1), regarding the claim that task representation shift is a "major issue", the experimental results do not at the moment provide evidence for this as claimed in the paper. Statistical significance analysis of the results should provide further information.

The Assumptions 4.7, 4.8, and 4.9 need to be motivated in more detail. In particular, in Assumption 4.9, assuming that fitting error decreases inversely proportional to the number of samples is questionable.

vol(Z) needs to be explained.

Other technical details, discussed below, need to be described in more detail.

DETAILS:

In more detail, the paper says "As shown by Figure 2, even with minor changes to the algorithms, the performance improvements are substantial." but that the performance improvements are "substantial" is not at all obvious but rather a result of random chance.
Statistical significance testing is needed to draw conclusions about the experimental results. This applies to all the results.

In "vol(Z) as the volume of the task representation coverage simplex.", please define "coverage simplex". What exactly is it? Furthermore, on lines 860 - 863, Lemma 8.2 is used. However, this Lemma applies in the case of discrete values and uses the number of values '|A|' in '2^|A|' but here '|A|' is replaced with vol(Z) resulting in 2^vol(Z). Why can this be done? This needs detailed explanation.

In Definition 4.5, I do not understand what "among the expectation of tasks before update of the context encoder and the policy" and "among the expectation of tasks after update of the context encoder and the policy." mean. Maybe the word "among" is here confusing. Can you provide a more explicit detailed definition? Just provide the equations for J^1(\theta_1) and J^2(\theta_2)?

On Line 281, the paper refers to Eq. (32) which has not been introduced yet.

Notation: using multiple characters to denote quantities such as "bs", "acc" etc. is not a good way. If this kind of textual description for variables is desired, one way is to use something like N_{\text{acc}}.

The notation is slightly confusing. Now \theta denotes policy parameters but it would be good to add also a symbol to denote the context encoder parameters to distinguish clearly in the equations what is being optimized.


RELATED WORK:

In "While ContrBAR (Choshen & Tamar, 2023) also benefits from the performance improvement guarantee in the OMRL setting, it is specifically served for its own framework", what does "it is specifically served for its own framework" mean? Since the proofs for performance improvement guarantee are one of the claimed contributions in this paper it is important to describe this in sufficient detail and discuss the differences.


LANGUAGE/PRESENTATION:

In "weakens the condition necessary for monotonic performance improvements, and may therefore violate the monotonicity.", rephrasing may be needed. "weakens" is slightly misleading since the sufficient conditions are actually stricter, not weaker, that is, when taking the task distribution shift into account larger policy improvements are needed to satisty monotonicity according to the analysis in this paper.

The sentence "However, ours considers the variation of task representation ignored by the previous training framework by imposing the extra condition to determine whether needs to update the context encoder." is missing some words.

On line 261: "As shown in Corollary 4.4, the monotonic performance improvement can be guaranteed with only better approximation to Z^*(·|x).": I maybe understand the intention of the text here but would be good to describe this more explicitly, that is, that Z(·|x) should be close to Z^*(·|x) such that the lower bound is small enough for finding a policy that improves on the old policy?

"we need to find the positive C to improve the performance monotonically." ->
"we need to find a positive C to improve the performance monotonically."

"Center around this motivation" -> "Centering around this motivation"

"an policy learning algorithm" -> "a policy learning algorithm"

**Questions:**

Update: I am happy with most of the authors' answers and recommending accepting the paper. I raised the rating to a strong 6.

Old questions:

1) Statistical significance analysis should be run in all experiments or claims on experimental results changed significantly, 2) Please answer all technical questions and comments.

---

> ### Author Response · Authors · 2024-11-19
> **Rebuttal by the authors**
>
> Dear Reviewer jUhN,
>
> We sincerely thank the reviewer for the constructive comments and the appreciation of our work. We hope our explanation below would clarify your concerns:
>
> $\textbf{Q1.Statistical significance analysis}$
>
> Please refer to the general response Q1.
>
> $\textbf{Q2.The details w.r.t the Assumptions}$
>
> Please refer to the general response Q2.
>
> $\textbf{Q3.Vol(Z) and coverage simplex}$
>
> We thank the reviewer for this valuable feedback!
> “Coverage simplex”[1] represents a geometric structure that encompasses all possible task representation Z, each task representation can be seen as a point within this simplex.
>
> We use the volume vol(Z) of this simplex to quantify the span of task representations.
> Here, we use vol(Z) in place of |A| as we are treating vol(Z) as an analog to the cardinality.
> According to our Assumption 4.8, the task representation space is discrete and limited, hence, using the cardinality of the task representation space to replace |A| is natural.
>
> To avoid confusion, we use |Z| to replace the notation of vol(Z), and clarify the meaning of |Z| in our Theorem 4.10.
> This term is treated as a constant within our theoretical framework and does not affect the algorithm. Nevertheless, if you have suggestions for a better notation, we would greatly appreciate your feedback.
>
> [1] An Information Geometry of Statistical Manifold Learning
>
> $\textbf{Q4.Definition 4.5: a more detailed definition}$
>
> “Among the expectation of tasks” denotes the outermost expectation over tasks, namely $\mathbb{E}_m$. To avoid potential confusion, we remove the expression w.r.t “Among the expectation of tasks” from our paper and use precise mathematical formulation instead. Additionally, we update the statement of Definition 4.5.
>
> $\textbf{Q5.Eq. (32) has not been introduced yet}$
>
> We thank the reviewer for pointing out this typo and we update our submission accordingly.
>
> $\textbf{Q6.Notation: Using multiple characters to denote quantities}$
>
> We thank the reviewer for this nice advice and we update our statement in Section 4.3 and the algorithmic box accordingly.
>
> $\textbf{Q7.The notation is slightly confusing. Now $\theta$ denotes policy parameters but it is good to add a symbol to denote the context encoder.}$
>
> We thank the reviewer for this feedback. We update the notation of the task representation as $Z(\cdot|x;\phi)$, where $\phi$ is the parameter of the context encoder, and accordingly update the whole paper. We also update the notation in Algorithm 1 from $q_\phi$ to $Z_\phi$ to avoid confusion.
>
> $\textbf{Q8.Related works}$
>
> We add more details of the ContraBAR in our related works to highlight two points: 1) the theoretical insight of ContraBAR lies in the online setting while ours lies in the offline setting, and 2) ContraBAR optimizes a specific approximate bound of $I(Z;M)$, namely $I(z;s_{t+1},r_t|a)$, and set an assumption on this approximation while we face a large class of algorithms that optimize various approximate bounds of $I(Z;M)$.
>
> To further highlight the core claim of our paper, we move the full related work into appendix and leave the COMRL and performance improvement guarantee parts.
>
> $\textbf{Q9.Weaken the condition necessary for monotonic performance improvement...}$
>
> We thank the reviewer for pointing out this misleading expression. Here, we use “weaken” to point out that the previous theoretical framework does not account for the variation of the task representation, making the condition for monotonic performance improvement insufficient.
>
> In fact, $\textbf{the task representation varies}$. Hence, the previous framework can not truly capture the condition for monotonic performance improvement (weak).
>
> To avoid this confusion, we rephrase our sentence as: “Furthermore, after scrutinizing this optimization framework, we observe that the condition for monotonic performance improvements does not consider the variation of the task representation. When these variations are considered, the previously established condition may no longer be sufficient to ensure monotonicity, thereby impairing the optimization process.”
>
> $\textbf{Q10.However, ours considers the variation of task representation ignored by the previous training framework,...}$
>
> We rephrase this sentence: However, our training framework considers the previously ignored variation of task representation by introducing an extra condition to decide whether the context encoder should be updated.
>
> $\textbf{Q11.As shown in Corollary 4.4, the monotonic performance improvement can be guaranteed with only better approximation.}$
>
> We thank the reviewer for your advice! We rephrase this sentence: $Z(\cdot|x;\phi)$ should be close to $Z(\cdot|x;\phi^*)$ such that the lower bound is small enough for finding a policy to achieve monotonic performance improvement.
>
> $\textbf{Q12.Other Typos}$
>
> We thank the reviewer for pointing out these typos, and we update our paper to “find a positive C”, “centering around”, “a policy learning algorithm”.

---

> > ### Comment · Reviewer_jUhN · 2024-11-25
> >
> > I am happy with most of the answers. I raised the rating to a strong 6.

---

> > > ### Author Response · Authors · 2024-11-25
> > > **Thank You**
> > >
> > > Dear Reviewer jUhN,
> > >
> > > Thank you for updating the score, and we sincerely appreciate your suggestions, which have greatly helped us improve our paper.
> > >
> > > Best,
> > >
> > > All authors

---

> ### Author Response · Authors · 2024-11-25
> **Official Comment by Authors**
>
> Dear Reviewer jUhN,
>
> We would like to express our sincere gratitude to you for reviewing our paper and providing valuable feedback. We have tried our best to respond to your concerns/questions and hope your concerns can be potentially addressed.
>
> Notably, given that we are approaching the deadline for the discussion phase, please feel free to reach out if you have any additional questions or require further clarification.
>
> Best,
>
> All authors

---

### Official Review · Reviewer_FSsk · 2024-11-02

**Soundness:** 2
**Presentation:** 2
**Contribution:** 2
**Rating:** 6
**Confidence:** 4

**Summary:**

This paper explains the reasons for the effectiveness of previous context-based offline meta reinforcement learning methods and introduces a task representation shift problem. The authors further demonstrate both theoretically and empirically that improving the update process of the context encoder for this problem can significantly enhance the performance of the original COMRL algorithm.

**Strengths:**

1.Provides a new perspective for learning better task representations in COMRL.
2.The paper includes a detailed mathematical derivation process.
3.The authors provide their code implementation in the supplementary materials.

**Weaknesses:**

Experimental Section:
1. The experimental results are obtained only within the improved FOCAL framework, lacking experiments on important baselines such as CORRO [1], CSRO [2]. I believe conducting experiments in just one framework is insufficient to verify effectiveness. It would be more convincing if significant results could be demonstrated across multiple baselines.
2. The authors only consider four environments, while in CSRO and UNICORN, each algorithm considers six environments under Mujoco.

**Questions:**

1. The authors' description of the task representation shift is unclear. Can it be understood as the phenomenon where the task representation deteriorates after updating the encoder? A specific example would be helpful for better understanding this point.
2. The authors introduce a large number of assumptions in Section 4. Has the reasonableness of these assumptions been justified?
3. When choosing the Ant environment for experiments under Mujoco, was it randomly selected, or was it only the environment that showed the best results?
4. How are the hyperparameters selected for different environments? It seems from the results in Figure 2 that the experimental outcomes are quite sensitive to the choice of k/acc. Is it possible to analyze the results of different hyperparameters in relation to the characteristics of the environments themselves?

---

> ### Author Response · Authors · 2024-11-19
> **Rebuttal by the authors**
>
> Dear Reviewer FSsk,
>
> We sincerely thank the reviewer for the constructive comments and the appreciation of our work. Our response to your concerns/questions:
>
> $\textbf{Q1. The description of the task representation shift}$
>
> We thank the reviewer for the feedback! There might be some misunderstandings regarding our method. Our method does not aim to learn a better task representation but focuses on how to adjust the learning process of the task representation based on previous COMRL methods.
>
> According to our theory, previous works suggest that the condition for monotonic performance improvements is shown in Eq. (7). However, $\textbf{they ignore the fact that the task representation also varies during the optimization process}$. If we take this variation into consideration, the condition for monotonic performance improvement becomes Eq. (10). This theoretical insight highlights the importance of reining in the task representation variation, specifically $|Z_2-Z_1|$. Without such control, it is more likely the condition necessary for performance improvement would be violated.
>
> For example, if we assume that $Z_1$ is trained from scratch and the update process from $Z_1$ to $Z_2$ brings $Z_2$ close to $Z^*$, then with the condition in Eq. (7), the monotonic performance improvement can be easily achieved with small $\epsilon^*_{12}$.
> However, for the condition in Eq. (10), small $\epsilon^*_{12}$ may cause the violation of monotonicity as $|Z_2-Z_1|$ remains large.
>
> Based on this, our algorithm framework points out that to get better performance improvement, we need to consider two aspects to rein in the task representation shift: 1) When to update the context encoder and 2) How many times to update the context encoder when it needs to be updated.
>
> $\textbf{Q2.The experimental results are obtained within the improved FOCAL framework, lacking experiments on important baselines}$
>
> In our experiment, $\textbf{we are not limited to be within the FOCAL}$, which is the upper bound of $I(Z;M)$.
>
> We cover the reconstruction, which is the lower bound of $I(Z;M)$ and $\textbf{equivalent to the objective of CORRO in the offline setting}$ (please see theoretical details in UNICORN [1]). We also cover the cross-entropy, which is the direct approximation towards $I(Z;M)$ (please refer to Appendix 8.4). $\textbf{These two algorithmic backbones have nothing to do with FOCAL at the algorithmic implementation level}$.
>
> To avoid potential confusion, we update the statement in Section 4.3 and the last paragraph of the Introduction to make it clearer.
>
> We appreciate your suggestions to expand the baselines and we include CSRO as an addition since it linear interpolates between the upper-bound and the lower-bound. We believe it indeed helps us to make our results more convincing. Thank you again for your insights and encouragement to strengthen our work.
>
> $\textbf{Q3.The authors only consider 4 environments}$
>
> Please refer to the general response Q1.
>
> $\textbf{Q4.Large number of assumptions}$
>
> Please refer to the general response Q2.
>
> $\textbf{Q5.When choosing the Ant environment for experiments under Mujoco, was it randomly selected?}$
>
> Based on the performance reported in UNICORN [1], we observe that, among the five MuJoCo environments, $\textbf{only Ant-Dir and Walker-Param demonstrate a noticeable performance differentiation}$. Therefore, in our work, we randomly select Ant-Dir for MuJoCo benchmark. For the other three environments, we choose more complex settings from MetaWorld to better validate our theory. Additionally, $\textbf{we include Walker-Param as a newly added environment}$, where our experimental results also show considerable performance.
>
> $\textbf{Q6. How are the hyper-parameters for different environments? Is it possible to analyze the results of different hyper-parameters}$
>
> Please refer to the general response Q4.
>
> [1] Towards an Information Theoretic Framework of Context-Based Offline Meta-Reinforcement Learning

---

> > ### Comment · Reviewer_FSsk · 2024-11-25
> > **Response to the Rebuttal by the authors**
> >
> > I am satisfied with the majority of the answers and have increased the rating to a solid 6.

---

> > > ### Author Response · Authors · 2024-11-25
> > > **Thank You**
> > >
> > > Dear Reviewer FSsk,
> > >
> > > Thank you for updating the score, and we sincerely appreciate your suggestions, which have greatly helped us improve our paper.
> > >
> > > Best,
> > >
> > > All authors

---

> ### Author Response · Authors · 2024-11-25
> **Official Comment by Authors**
>
> Dear Reviewer FSsk,
>
> We would like to express our sincere gratitude to you for reviewing our paper and providing valuable feedback. We have tried our best to respond to your concerns/questions and hope your concerns can be potentially addressed.
>
> Notably, given that we are approaching the deadline for the discussion phase, please feel free to reach out if you have any additional questions or require further clarification.
>
> Best,
>
> All authors

---

### Official Review · Reviewer_SUPu · 2024-11-03

**Soundness:** 3
**Presentation:** 3
**Contribution:** 3
**Rating:** 8
**Confidence:** 3

**Summary:**

This paper addresses the overlooked issue of task representation shift in offline meta-reinforcement learning (OMRL), which can prevent consistent performance improvement.

The authors provide theoretical insights and practical strategies to control this shift, demonstrating that such control enables more stable and higher asymptotic performance. They propose adjustments in training, such as tuning batch sizes and accumulation steps, to manage task representation effectively. Experimental results across benchmarks validate that these adjustments lead to performance gains, highlighting the importance of considering task representation dynamics in OMRL.

Overall, this is a very interesting paper with great potential to inspire future work. I would be willing to increase the score if the authors could address my following concerns.

**Strengths:**

1. The paper is well-structured and clear.
2. The authors identified a unique challenge existing in offline meta-RL, task representation shift, which is highly novel.
3. The proof provided is detailed and logically rigorous, highlighting a flaw overlooked by previous work: it ignores the variation of task representation in the optimization process. I believe this is the most significant contribution of this paper.
4. The paper concludes with some interesting discussions, which have the potential to motivate future research.

**Weaknesses:**

1. The algorithm box does not clearly explain how $k$ and $acc$ are utilized. Adding a brief explanation in the red-highlighted part of the algorithm box about how these are calculated would make the algorithm more understandable.
2. The experiments are limited and need improvement; it would be beneficial to verify the impact of task representation shift in more diverse testing scenarios.
3. The results largely depend on the settings of hyper-parameters, such as $k$ and $acc$, which seem to be unstable. The paper lacks an analysis of the experimental effects caused by adjusting these two parameters.

**Questions:**

1. The setup in Section 4.3 is a bit confusing. Could the authors clarify what "accumulation steps of task representation shift" refers to? Also, does setting $k = 2 \times bs$ refer to the initial value for training?
2. It would be better to explain in the appendix how the cross-entropy-based loss can replace the loss in FOCAL, preferably by providing the corresponding expression.
3. The motivation for using the cross-entropy-based algorithm is somewhat unclear. Could you explain why it replaces the distance metric learning loss?
4. From the experimental results, the improvement seems marginal. Could you analyze the reasons behind this?
5. The authors state in section 6.2, "We recognize that the visualization result can be seen as an auxiliary metric to assist in determining the task representation." Why is the visualization result insufficient to fully represent the true task representation? The visualized convergence results being imperfect yet leading to better outcomes seem counterintuitive.

**Details Of Ethics Concerns:**

None.

---

> ### Author Response · Authors · 2024-11-19
> **Rebuttal by the authors**
>
> Dear Reviewer SUPu,
>
> We sincerely thank the reviewer for the constructive comments and the appreciation of our work. Our response to your concerns/questions:
>
> $\textbf{Q1. The algorithm box about $k$ and $acc$ and the confusing setup in Section 4.3.}$
>
> We sincerely thank the reviewer for pointing out the confusion in Section 4.3. The purpose of Section 4.3 is to explain the ways we use to rein in the task representation shift. According to our algorithmic box, the way to control the task representation shift can be seen as two aspects. 1) When to update the context encoder and 2) How many times to update the context encoder.
>
> Thus, to cover these two aspects, we introduce two parameters $k$ and $acc$. Here, $k=n×bs$ denotes that the context encoder needs to be updated every $n$ updates of the policy (according to our theory, this means $\epsilon^*_{12}$ accumulates $n$ times to meet the condition for updating the context encoder) and $acc=n$ denotes that when the context encoder needs to be updated, it is updated $n$ times. Please note that this parameter $k$ is distinct from $k$ used in our theoretical framework.
>
> To clarify, we update the statement in Section 4.3, the algorithmic box and update the notations in the experiment correspondingly.
>
> $\textbf{Q2. The experiments are limited and need improvement}$
>
> Please refer to our general response Q1.
>
> $\textbf{Q3. The results largely depend on the settings of hyper-parameters and lacks of analysis of the experimental effects.}$
>
> Please refer to our general response Q4.
>
> $\textbf{Q4. The cross-entropy-based loss}$
>
> We add a description w.r.t the cross-entropy-based objective in the Appendix. Please refer to Appendix 8.4 for more details.
>
> $\textbf{Q5. The improvements seem marginal}$
>
> According to our paired t-test results in Table 3, there always exist instances where the p-value is less than 0.05, indicating statistical significance. This suggests that even simple modifications can have a meaningful impact.
>
> Also, as shown in Appendix 8.6, if we use the evaluation performance to guide the learning of the context encoder (where the calculation of $k$ is determined by our theoretical framework), the performance improvement can get further enhancement. Hence, we also acknowledge the need for smarter algorithms to achieve stronger performance improvements (which has been stated in our limitation).
>
> Additionally, the final performance may also be restricted by the offline dataset. For example, on the Ant-Dir dataset, the average return is only 21, which may limit the extent of achievable improvements.
>
> $\textbf{Q6. Concerns in Section 6.2}$
>
> This is a good question. We would like to state that while visualization results produced by t-SNE can provide valuable insights into the quality of the learned task representation, they may not fully capture the performance of the downstream policy.
>
> Our theoretical analysis suggests that achieving better performance improvement involves not only optimizing Z towards the desired target $\textbf{but also effectively reining in the task representation shift}$. If the task representation shift is not properly adjusted, it may impede the attainment of the performance.
> We believe this highlights a crucial aspect of our work: the importance of both optimizing the task representation and reining in the task representation shift.

---

> > ### Comment · Reviewer_SUPu · 2024-11-24
> > **Follow-up Questions**
> >
> > Dear authors,
> >
> > I greatly appreciate the authors' detailed responses. I believe this is a very solid piece of work. However, the experimental section is still quite limited compared to other baseline works, and determining some key parameters through heuristics seems overly simplistic. Here are some follow-up questions based on the answers.
> > 1. The mitigation of the task representation shift issue led to only marginal performance improvements in most experiments, even with some additional strategies introduced in the Appendix. Is this because this task representation shift issue inherently cannot bring substantial gains to the offline meta-RL field, or because the proposed solution is limited?
> > 2. For Question 6, I think the authors' explanation is somewhat unconvincing. Better performance improvement indeed involves not only optimizing $Z$ towards the desired target but also effectively managing the task distribution shift. However, when the final performance converges, the learned task representation should reflect the differentiation between tasks. I believe this is a key assumption and motivation behind the context-based meta-RL framework. The explanation that "less desirable differentiation results lead to better performance" is likely due to the learned policy being suboptimal. Could the authors provide further explanation if my understanding is incorrect?

---

> ### Author Response · Authors · 2024-11-25
> **Rebuttal for the follow-up questions by the authors**
>
> Dear Reviewer SUPu,
>
> We thank the reviewer for the feedback! Our response to your concerns/questions:
>
> $\textbf{Q1. Determining some key parameters through heuristics seems simple and lead to only marginal performance improvements.}$
>
> We agree with the reviewer that we adopt a simple strategy to show the potential of reining in task representation shift (which we have stated in our limitation) since $\textbf{we do not claim to solve this issue by providing a strong algorithm, but stand for proposing this issue}$.
>
> We kindly remind the reviewer to focus on the pre-training scheme, $\textbf{which is the case where task representation shift is completely ignored}$.
> When comparing the pre-training with the best case of reining in the task representation shift, the performance improvement is no longer marginal, e.g. the mean of pre-training cross-entropy is near 200 while the mean of the best condition in our heuristic setting is near 300 (50% improvement); the mean of pre-training contrastive is near 175 while the mean of the best is near 265 (50% improvement),...,
>
> The reason for less performance gains against the original is that $\textbf{$N_{k}=1, N_{acc}=1$ can indeed be seen as a way to rein in the task representation shift.}$
>
> Nevertheless, we also invite the reviewer to see Table 3 in Appendix 8.4, the results of the paired t-test can also support that the performance improvement against the original setting has statistical significance, thereby $\textbf{achieving our goal to be a starting point for future research}$.
>
> And, as we use the offline dataset provided by UNICORN, according to the performance reported in UNICORN, the SOTA algorithms $\textbf{after carefully sweeping the parameters}$ namely UNICORN and CSRO, achieve the best performance 276 on Ant-Dir, 407 on Walker-Param, 2774 on Reach, and the original FOCAL only achieves 217 on Ant-Dir, 308 on Walker-Param, 2423 on Reach.
> By using the simple heuristic strategy, $\textbf{some previously weak baselines can beat the SOTA}$, e.g. cross-entropy can achieve 291 on Ant-Dir, FOCAL can achieve 450 on Walker-Param, FOCAL can achieve 2802 on Reach.
> $\textbf{From the view of simplifying the algorithm and reducing the effort for parameter-tuning, the task representation shift is also valuable.}$
>
> Since our main focus is to propose the task representation shift issue, our contributions in theory are $\textbf{three folds}$. Note that $\textbf{the baseline works like FOCAL, CORRO, and CSRO do not hold such strong theoretical contributions}$:
>
> (1) Provide a performance improvement guarantee for previous COMRL methods, $\textbf{which is the first in the offline setting}$.
>
> (2) Consider the variation of task representation explicitly and refine the condition for monotonic performance improvement guarantee.
>
> (3) Give theoretical proof of how we can achieve monotonic performance improvement guarantee.
>
> Additionally,  the experiments conducted in previous works have three main components: baseline comparison, ablation and visualization.
>
> In our work, we $\textbf{integrate the baseline comparison and ablations}$ (if we view the heuristic strategy as the hyper-parameter) into Section 5.1, since our framework builds upon these previous algorithms.
> To extend the applicability of our theory, we also add an experiment on different-quality datasets, which is shown in Section 5.2.
> We also add discussions $\textbf{concerning visualization of the task representation}$ in our Discussion.
>
> Hence, compared to other baseline works, our experimental design is reasonable and can validate our claims.

---

> ### Author Response · Authors · 2024-11-26
> **Rebuttal for the follow-up questions by the authors**
>
> $\textbf{Q2. The concerns for question 6}$
>
> We can use another example to explain this further.
> The policy has converged at 300K training steps.
> Hence, we use the example that pre-training cross-entropy at 300K v.s. the reconstruction for $N_{k}=3,N_{acc}=1$ at 300K.
> Though the cross-entropy demonstrates better visualization results, the final performance for pre-training cross-entropy is less than the reconstruction for $N_{k}=3,N_{acc}=1$, which reins in the task representation shift.
> Hence, $\textbf{ignoring task representation shift would cause an effect that “less desirable differentiation results lead to better performance”}$.
> Based on this effect, we aim to claim that using the visualization results to $\textbf{imply the final performance}$ is unreliable.
> We agree with the reviewer that the reason behind this (even the performance converges) may be the sub-optimal learned policy as ignoring the task representation shift would impede policy learning.
>
>
> If we go further, and both algorithms being compared take task representation shift into account, we also think it is not certain that better differentiation would lead to better performance.
> We encourage the reviewer to see the visualization results of UNICORN[1] in its Appendix C.3.
> Although FOCAL demonstrates better visualization results, its final performance still falls behind UNICORN.
> We speculate that excessive differentiation may lead to a failure to capture the similarities between tasks.
> Nevertheless, for this point, we are open to further discussions.
>
> In general, the visualization results $\textbf{may}$ represent the true task distribution, however, it cannot sufficiently imply the final performance.
>
> To avoid confusion, we update our statement in Section 6.2 "Hence, it is insufficient to imply performance based on such evaluation principles.Nevertheless, we recognize that the visualization result can be seen as an auxiliary metric to assist in determining the task representation." to "Hence, the visualization results may represent the true task distribution but cannot sufficiently imply the final performance." and also update the example in Figure 5.
>
> [1] Towards an Information Theoretic Framework of Context-Based Offline Meta-Reinforcement Learning.
>
> ---
> We hope our response could potentially address your concerns/questions. If you need any further elaboration, please feel free to reach out!

---

> > ### Comment · Reviewer_SUPu · 2024-11-26
> >
> > Dear authors,
> >
> > These responses well resolved my previous concerns. Thus I would like to raise my final rating to 8.

---

> > > ### Author Response · Authors · 2024-11-27
> > > **Thank You**
> > >
> > > Dear Reviewer SUPu,
> > >
> > > Thank you for updating the score, and we sincerely appreciate your suggestions, which have greatly helped us improve our paper.
> > >
> > > Best,
> > >
> > > All authors

---

### Official Review · Reviewer_FVd6 · 2024-11-04

**Soundness:** 3
**Presentation:** 3
**Contribution:** 3
**Rating:** 8
**Confidence:** 3

**Summary:**

This work attempts to explain the performance improvement of the Offline Meta RL (ORML) optimization framework. It identifies that the variation of task representation learned through the optimization process if often ignored. Such issue violates the condition of monotonic performance improvements. Thus, addressing task representation shift with carefully designed encoder updates is necessary. Experimental results across two different benchmarks with 3 different types of training objective and data qualities have been presented.

**Strengths:**

1. This work considers a nuanced issue in traditional OMRL optimization framework. It critically investigates the components and show that variation in task representation is fundamental to monotonic performance improvement. It introduces the phenomena called "task representation shift".
2. Theoretical justification for the claims are well presented and it outlined a complete algorithmic framework.
3. It presents rigorous experimental validation using several environments from two different benchmarks with 3 types of objective functions and 3 types of data sets.

**Weaknesses:**

1. The task variation used in the experiments could be improved by including more distinct tasks.
2. Lack of results with more statistically significant metrics such as interquartile mean with confidence interval. Such comparison would help the reader as the general standard deviation highly overlaps.

**Questions:**

Can you elaborate more on how different batch sizes induce different task level contexts?

---

> ### Author Response · Authors · 2024-11-19
> **Rebuttal by the authors**
>
> Dear Reviewer FVd6,
>
> We sincerely thank the reviewer for the constructive comments and the appreciation of our work. Our response to your concerns/questions:
>
> $\textbf{Q1. The task variation used in the experiments could be improved by including more distinct tasks. / Lack of results with more statistically significant metrics}$
>
> Please refer to the general response Q1.
>
> $\textbf{Q2. Can you elaborate more on how different batch sizes induce different task level contexts?}$
>
> Thank you for the question. We’re not entirely clear on what you mean by “different task-level context”. Nevertheless, regarding the different batch sizes, we do not need to tune the training batch size of the context encoder. In our approach, we follow the settings in UNICORN [1], where we randomly select a trajectory and use all transitions from that trajectory as our batch to train the context encoder. This keeps our batch size fixed for each update.
>
> According to our proposed theory, consistently updating the policy allows for increasing $\epsilon^*_{12}$, which in turn decreases $k$. Therefore, we only update the context encoder when $k$ is less than our given batch size, eliminating the need to adjust the context encoder's training batch size. Notably, if the training batch size is too small, it may insufficiently guide the downstream policy learning, but our update mechanism can effectively avoid this issue. We hope this can potentially address your question. If that’s not the case, could you kindly clarify so we can provide more detailed information?
>
> [1] Towards an Information Theoretic Framework of Context-Based Offline Meta-Reinforcement Learning

---

> > ### Comment · Reviewer_FVd6 · 2024-12-03
> > **Reply to the Authors**
> >
> > Dear Authors,
> >
> > Thank you for your response and for adding the new results as part of the Q1.
> >
> > Thanks for adding more information between lines 321-350. This makes the algorithm much easier to follow. "use all transitions from that trajectory as our batch to train the context encoder" - this helps to clear my confusion. Also, your reply to the author SUPu in Q1 - "Please note that this parameter $k$ is distinct from $k$ used in our theoretical framework" - is helpful in resolving my wrong impression. I would suggest using two different parameters to avoid confusion.
> >
> > I would keep my positive score.

---

> > > ### Author Response · Authors · 2024-12-03
> > > **Thank You**
> > >
> > > Dear Reviewer FVd6,
> > >
> > > Thank you for your suggestions and in our updated submission, we have used $N_k$ to differentiate these two parameters.
> > >
> > > Thank you again for your high recognition for our work.
> > >
> > > Best,
> > >
> > > All authors

---

### Author Response · Authors · 2024-11-19
**Global response by the authors**

We sincerely thank the reviewers, ACs and PCs for ensuring high-quality review of the paper. We find all reviews constructive and helpful for making our paper stronger.

Here we summarize some key/common questions raised and provide our general response as follows:

$\textbf{Q1. Lack of experiments and statistically significant analysis}$

We are grateful to the reviewer for proposing this key issue. Correspondingly, to make our claim more convincing, we add a new algorithmic baseline CSRO [1], which uses linear interpolation between the lower bound and the upper bound to better approximate $I(Z;M)$, and add two challenging benchmarks Walker-Param and Push.

To better demonstrate our performance improvement, we do a thorough statistically significant analysis by using paired t-test and report p-value.  We also report the mean and standard deviation for all cases in our experiment section.

Please refer to Table 3 in Appendix 8.4.

$\textbf{Q2. Justification of Assumption 4.7, 4.8, 4.9}$

We provide a thorough discussion and justification of these three assumptions in Appendix 8.3. Please refer to Appendix 8.3 for more details.

$\textbf{Q3. The confusion in our experimental setting Section 4.3}$

To avoid potential confusion, we rephrase Section 4.3 and the Algorithm box.
We also update a more concise and clearer description of the algorithms used in this paper in Section 4.3 and the last paragraph of the Introduction, e.g. to emphasize the relationship with $I(Z;M)$, in the hope of more accurately highlighting that the situations covered in this paper are general.

$\textbf{Q4. Clarification of hyper-parameter selecting}$

Table 3 shows that settings with $N_{\text{k}} > 1, N_{\text{acc}}=1$ achieve better performance more frequently than those with $N_{\text{k}} = 1, N_{\text{acc}}>1$. Thus, we recommend prioritizing adjustments to $N_{\text{k}}$ during parameter tuning, as it not only improves performance but also reduces training time. In contrast, tuning $N_{\text{acc}}$ increases training time due to additional context encoder updates per alternating step. (We add this description in our updated Section 5.2.)

Our work theoretically highlights the ignored issue of task representation shift and demonstrates through experiments that reining in this shift, even with simple adjustments, consistently improves performance compared to the original settings. We hope this serves as a starting point to encourage further exploration of this issue.

We have acknowledged the need for smarter algorithms with low-sensitive hyperparameters as a limitation and future direction. To extend our work, we conduct an additional experiment on Ant-Dir, using a $\textbf{fixed hyperparameter}$ across three algorithms. By deciding context encoder updates based on the relationship between policy improvement and $k$ in Eq.(11), we observe performance improvement can get a further enhancement(please refer to Appendix 8.6 for more details). We hope this theory can potentially inspire the development of more stable algorithms in the future.

[1] Context Shift Reduction for Offline Meta-Reinforcement Learning.

---

### Meta-Review · Area_Chair_2AEe · 2024-12-10

**Metareview:**

This work investigates task representation shift in offline meta reinforcement learning, providing theoretical insights and practical strategies to control this shift, such as tuning batch sizes and accumulation steps.  Experiments are well designed and executed, across two different benchmarks with three different types of training objective and data qualities, validating the proposed methods. The reviewer believes that this is an important but overlooked issue by the meta RL community and this paper is a solid work in identifying this issue and proposing the first working solutions. All reviewers agree that this is a good work and should be accepted and I agree with the reviewers' evaluation and believe this work could open up an interesting future research direction.

**Additional Comments On Reviewer Discussion:**

The reviewers raised some questions which are subsequently addressed by the authors. At the end of the rebuttal period, all reviewers unanimously agree that this is a good work and should be accepted.

---

### Decision · Program_Chairs · 2025-01-22

Accept (Poster)